# RETHINKING THE ROLE OF DYNAMIC SPARSE TRAINING FOR SCALABLE DEEP REINFORCEMENT LEARNING

## ABSTRACT

Scaling neural networks has driven breakthrough advances in machine learning, yet this paradigm fails in deep reinforcement learning (DRL), where larger models often degrade performance due to unique optimization pathologies such as plasticity loss. While recent works show that dynamically adapting network topology during training can mitigate these issues, existing studies have three critical limitations: (1) applying uniform dynamic training strategies across all modules despite encoder, critic, and actor following distinct learning paradigms, (2) focusing evaluation on basic architectures without clarifying the relative importance and interaction between dynamic training and architectural improvements, and (3) lacking systematic comparison between different dynamic approaches including sparse-to-sparse, dense-to-sparse, and sparse-to-dense. Through comprehensive investigation across modules and architectures, we reveal that dynamic sparse training strategies provide module-specific benefits that complement the primary scalability foundation established by architectural improvements. We finally distill these insights into Module-Specific Training (MST), a practical framework that further exploits the benefits of architectural improvements and demonstrates substantial scalability gains across diverse RL algorithms without algorithmic modifications.

## 1 INTRODUCTION

Scaling neural networks has emerged as a fundamental driver of progress in modern machine learning, with larger models consistently delivering superior performance. This fundamental scaling paradigm, however, fails dramatically in deep reinforcement learning (DRL), where increased model size frequently degrades rather than improves performance (Nauman et al., 2024a;b). This limited scalability of DRL models can be largely attributed to severe optimization pathologies that emerge during training (Nikishin, 2024; Nauman et al., 2024a). Primary pathological behaviors manifest as capacity collapse (Lyle et al., 2022) and plasticity loss (Nikishin et al., 2022; Sokar et al., 2023), whereby networks progressively lose their ability to learn from newly collected experiences, resulting in catastrophic learning inefficiency or even stagnation (Ma et al., 2024; D'Oro et al., 2023). Moreover, these phenomena intensify as model size increases, essentially limiting the scaling potential that has been driving advances in other domains like natural language processing and computer vision (Ceron et al., 2024b;a). Consequently, developing RL-tailored DL mechanisms and employing targeted interventions to prevent networks from falling into these pathological states has become necessary for achieving truly efficient and scalable DRL algorithms (Schwarzer et al., 2023; Nauman et al., 2024b).

Network architectures and training strategies form the core mechanisms of deep learning (DL) systems. Recent advances in addressing the unique pathologies in DRL have emerged along two complementary directions: architectural innovations that enhance DRL networks while maintaining conventional dense training, and training strategy improvements on top of standard network architectures (Klein et al., 2024). Over the past few years, both directions have advanced significantly and contributed essential insights toward DRL scalability. • Architectural improvements including better normalization (Bhatt et al., 2024; Lyle et al., 2024a), residual connections (Nauman et al., 2024b; Wang et al., 2025), and activation function variants (Lewandowski et al., 2025; Park et al., 2025) have each shown performance gains on their own. Recent approaches such as BRO (Nauman et al., 2024b) and Simba (Lee et al., 2024) combine these elements effectively, achieving strong scaling performance without modifying underlying RL algorithms. • For training strategies, dynamic alternatives to conventional static training have demonstrated significant benefits in DRL. Approaches

including dynamic sparse training[1] (Graesser et al., 2022; Tan et al., 2023; Arnob et al., 2024), dense-to-sparse pruning (Ceron et al., 2024a), and sparse-to-dense growth (Liu et al., 2024) differ in specific implementation but share a common insight: *introducing network sparsity and dynamicity during training helps DRL networks avoid optimization pathologies and achieve better scalability.*

Despite growing recognition of the potential benefits of dynamic sparse training in DRL, its widespread adoption remains limited due to three critical drawbacks in existing research. • **First**, existing work typically applies uniform dynamic training strategies across all network components, despite the fact that encoder, critic, and actor modules follow fundamentally different learning paradigms with distinct optimization characteristics (Ma et al., 2024; Lee et al., 2024). For example, pruning techniques that enhance scalability in value-based DRL fail when applied to actor-critic algorithms due to their distinct gradient dynamics (Ceron et al., 2024a). This evidence suggests that *dynamic training effectiveness should be evaluated separately for each module type*. • **Second**, evaluations of dynamic training have been conducted predominantly on basic architectures like vanilla MLPs, with limited investigation of their effects on recent advances in DRL network design (Liu et al., 2024; Graesser et al., 2022). While dynamic training methods demonstrate benefits for basic networks suffering from severe pathologies, their interaction with architectural improvements remains unclear: *which approach has greater impact on scalability, and whether dynamic training methods provide additional benefits beyond well-designed architectures*. • **Third**, there have been a severe lack of comparison between different dynamic training approaches, including sparse-to-sparse, dense-to-sparse, and sparse-to-dense paradigms. Although they all introduce network dynamicity, each creates distinct sparsity patterns throughout training that affect learning dynamics differently (Liu et al., 2022; Arnob et al., 2021). Hence, *a comprehensive comparison is crucial to provide practical dynamic sparse training implementation guidance for DRL algorithm design*.

To address these gaps, we conduct a systematic empirical investigation examining how dynamic training effectiveness varies across different DRL modules and architectural configurations. Specifically, we base our experiments and analysis on MR.Q (Fujimoto et al., 2025), which provides explicit decoupling of module training methods: encoders learn through self-supervised dynamics prediction, critics learn through temporal difference learning, and actors learn via policy gradients. To investigate architecture-training interactions, we implement six network variants with different combinations of normalization, activation functions, and residual connections (detailed in Section 2.2). Building on the RigL (Evci et al., 2020) framework that prunes connections by weight magnitude while growing new ones based on gradient information, we implement three dynamic training regimes: ◆ Dynamic Sparse Training (DST), ◀ Sparse-to-Dense (S2D), and ▶ Dense-to-Sparse (D2S), while also establishing ● Dense Training and ■ Static Sparse Training (SST) with one-shot pruning as baselines for fair comparison. Our findings clarify the effectiveness and limitations of dynamic sparse training, providing clear implementation guidance. Key insights include:

1. Interaction between DRL network architecture and training strategy:
   - Architectural improvements provide the primary foundation for DRL scalability, with dynamic training strategies offering complementary but secondary benefits.
   - Dynamic sparse training significantly helps basic networks suffering from pathologies, but provides diminishing returns when applied to architecturally advanced networks.
   - The combination of well-designed architecture and appropriate training strategies yields synergistic scalability benefits that neither approach can achieve alone.

2. Module-specific responses to dynamic sparse training, i.e, network sparsity and dynamicity:
   - Self-supervised encoder can maintain scalability with proper architecture, making dynamic sparse training unnecessary compared to simple dense configurations. (Section 3.1)
   - Critics suffer from severe plasticity loss even with advanced architecture, making dynamic sparse training consistently beneficial compared to static configurations. (Section 3.2)
   - Static sparse training with advanced architecture prevents actors from developing pathologies during scaling, avoiding dynamicity that disrupts policy stability. (Section 3.3)

Integrating these insights, we propose Module-Specific Training (MST) as a systematic framework that assigns tailored training strategies to each DRL module: dense training for the encoder with

---

[1]We use 'dynamic sparse training' as an umbrella term for methods that introduce sparsity and dynamically adapt network topology, including but not limited to the specific DST algorithm.

strong architectural foundations, DST for the critic to address TD learning pathologies, and SST for the actor to balance plasticity with policy stability. When paired with strong architectural foundations, MST enables successful scaling to massive 362M parameter models while maintaining stable learning dynamics and achieving superior sample efficiency. Crucially, MST accomplishes this without modifying underlying RL algorithms, and generalizes effectively across different algorithmic frameworks including SAC and DDPG, establishing a practical pathway toward truly scalable DRL.

## 2 PRELIMINARY

In this section, we establish the foundational elements for our investigation: dynamic sparse training regimes, network architecture variants, and decoupled module training. The comprehensive background on DRL optimization pathologies and their mitigation approaches is provided in Appendix A.

### 2.1 DYNAMIC SPARSE TRAINING REGIMES AND STATIC BASELINES

To comprehensively evaluate the impact of dynamic training, we implement three distinct dynamic training regimes, alongside static sparse training as a baseline to isolate sparsity effects. We outline key characteristics below, with implementation details and hyperparameters in Appendix B.1.

● **Dense Training**: The conventional approach where all network parameters are initialized and trained throughout the learning process, maintaining full connectivity between neurons.

■ **Static Sparse Training (SST)**: SST (Liu et al., 2022) applies one-shot random pruning at initialization to create a fixed sparse topology with predefined layer-wise sparsity. Following prior findings that *Erdős-Rényi (ER)* initialization outperforms uniform sparsity (Graesser et al., 2022; Tan et al., 2023; Ma et al., 2025), we adopt ER throughout our experiments.

◆ **Dynamic Sparse Training (DST)**: DST maintains constant sparsity while dynamically evolving topology during training. Starting from a randomly pruned network, we implement *RigL* (Evci et al., 2020), which prunes connections based on weight magnitude and grows new ones guided by gradient.

◀ **Sparse-to-Dense Training (S2D)**: S2D begins sparse and gradually increases connectivity, releasing new parameters to maintain learnability (Liu et al., 2024). We implement this by extending RigL's framework with a decreasing sparsity schedule from high initial values to zero.

▶ **Dense-to-Sparse Training (D2S)**: D2S progressively prunes connections starting from a dense network, potentially helping dormant neurons escape optimization traps (Ceron et al., 2024a). Our D2S implementation follows the RigL framework with a gradual magnitude-based pruning schedule.

### 2.2 NETWORK ARCHITECTURE ADVANCES TAILORED FOR DEEP RL

The DRL community traditionally treated neural networks primarily as function approximators, directing research efforts toward core RL challenges such as exploration (Ciosek et al., 2019) and value overestimation (Fujimoto et al., 2018) rather than network architecture design. Hence, for a long period, most DRL research defaulted to basic MLPs (Fujimoto et al., 2023), adding only a few convolutional layers when processing visual observations (Yarats et al., 2022). Only when network pathologies such as plasticity loss were recently identified as critical bottlenecks did research begin to focus on architectural improvements to address these unique issues.

Three architectural elements have emerged as particularly significant: ● **First**, after ReLU was identified as a key contributor to dormant neurons (Sokar et al., 2023), numerous activation function variants have been proposed to mitigate this pathology, such as CReLU (Abbas et al., 2023) and AID (Park et al., 2025). In this paper, we employ ELU (Clevert et al., 2015) as a representative improvement in this direction. ● **Second**, various normalization techniques have demonstrated effectiveness, including Spectral Normalization (Bjorck et al., 2021),

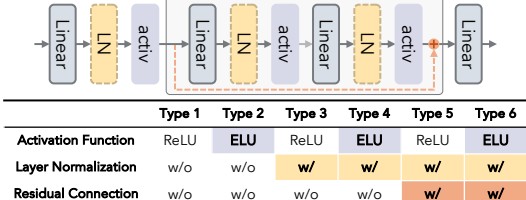

| | Type 1 | Type 2 | Type 3 | Type 4 | Type 5 | Type 6 |
|---|---|---|---|---|---|---|
| **Activation Function** | ReLU | ELU | ReLU | ELU | ReLU | ELU |
| **Layer Normalization** | w/o | w/o | w/ | w/ | w/ | w/ |
| **Residual Connection** | w/o | w/o | w/o | w/o | w/ | w/ |

Figure 1: Network architecture variants for our investigation, incorporating three key architectural improvements: activation functions, layer normalization, and residual connections.

Batch Normalization (Bhatt et al., 2024), Hyperspherical Normalization (Lee et al., 2025), and the widely adopted Layer Normalization (Lyle et al., 2023; 2024a). • **Third**, beyond merely using ResNet as a visual encoder (Espeholt et al., 2018), BRO (Nauman et al., 2024b) and Simba (Lee et al., 2024) have demonstrated that incorporating residual blocks is crucial for scalable policy and value networks. To investigate interactions between dynamic sparse training regimes and architectural design, we implement six network variants with progressive combinations of these elements, as shown in Figure 1. Detailed architectural designs and analyses can be found in Appendix B.2.

## 2.3 DECOUPLED MODULE TRAINING VIA SPECIFIC LEARNING PARADIGMS

To systematically investigate how distinct RL components respond to dynamic training, we adopt MR.Q (Fujimoto et al., 2025), which explicitly decouples training of the encoder, critic, and actor modules. Each module follows fundamentally different learning paradigms, resulting in unique optimization characteristics that demand tailored network properties: • **Encoder** learns through **self-supervised** dynamics prediction, where the loss is composed of three components: reward prediction, next-state dynamics prediction, and terminal state prediction (Fujimoto et al., 2025; 2023). By unrolling these signals over a short horizon, the encoder develops rich representations grounded in environment dynamics rather than non-stationary value targets, transforming diverse observations into a unified latent space (Schwarzer et al., 2021). • **Critic** trains through **temporal difference (TD) learning** with bootstrapped targets based on TD3 (Fujimoto et al., 2018). This paradigm is especially prone to plasticity loss (Ma et al., 2024; Sokar et al., 2023), wherein bootstrapping from non-stationary targets leads to unbounded parameter growth and eventual capacity collapse (Lyle et al., 2023). • **Actor** employs **deterministic policy gradient** to optimize parameters toward maximizing expected returns (Silver et al., 2014). This direct optimization approach differs from both self-supervised and TD learning, creating effective but potentially inflexible behaviors that depend on stable gradient pathways (Nauman et al., 2024a). Algorithm details are provided in Appendix B.3.

These fundamental differences in learning paradigms have led to numerous observed implementation distinctions across modules. For example, actor networks can tolerate significantly higher sparsity levels than critics when reducing computational budgets (Graesser et al., 2022), while scaling critic networks yields substantially greater benefits when pursuing RL scaling laws (Lee et al., 2024). This suggests value functions represent more complex mappings requiring greater capacity. In visual RL settings with all three modules, critics have been identified as suffering the most severe plasticity loss, becoming the primary performance bottleneck (Ma et al., 2024). Further evidence comes from contrastive RL (Eysenbach et al., 2022), which demonstrates markedly superior scalability when critics are trained with InfoNCE objectives rather than TD losses (Wang et al., 2025). Most relevant to our study, dense-to-sparse pruning methods that significantly enhance scalability in value-based RL (approximated as critic-only systems) fail to transfer successfully to actor-critic algorithms (Ceron et al., 2024a). These documented differences strongly motivate our approach of investigating the distinct impacts of different training regimes across the encoder, critic, and actor modules separately.

## 3 INVESTIGATION: HOW DST STRATEGIES IMPACT DEEP RL SCALABILITY?

To systematically understand how network dynamicity impacts deep RL scalability, we conduct a *module-specific investigation* following the natural dependency chain: encoders train independently, critics build on encoders, and actors depend on both. We progressively examine each module in isolation, comprehensively comparing five training strategies (● Dense, ■ SST, ◆ DST, ◄ S2D, ▶ D2S) and their *interactions with architectural variants* to reveal fundamental scaling principles.

**Setup.** We conduct our experiments using MR.Q (Fujimoto et al., 2025) on state-based *Dog Run* and *Humanoid Run*, widely recognized as the most challenging tasks in the DMC suite (Tassa et al., 2018) due to their complex dynamics and high-dimensional action spaces. For each module, we first examine the impact of different training strategies and architectural variants at default network sizes, then conduct scalability analysis by scaling networks by factors of 3× and 5× in both width and depth, creating models spanning three orders of magnitude in parameter count. We maintain a consistent sparsity level of 0.6 across all sparsity-based training regimes (60% of parameters pruned). Specifically, SST and DST maintain constant 0.6 sparsity throughout training, D2S gradually increases from 0 to 0.6, and S2D decreases from 0.6 to 0. Unless stated otherwise, all experimental results in this paper are averaged over 8 random seeds.

## 3.1 ENCODER: NETWORK ARCHITECTURE DOMINATES SELF-SUPERVISED LEARNING

We begin our investigation with the encoder that learns through self-supervised dynamics prediction. To isolate the effects of encoder design, we vary only encoder architecture and training regime, while maintaining the original MR.Q architecture and size for critic (Type 4) and actor (Type 3) networks with standard dense training, which ensures reliable and consistent operation.

**Comparison at Base Scale.** We evaluate six architecture types and five training regimes for encoders at default scale. Detailed comparison is provided in Appendix D due to space constraints. Figure 15 reveals two key findings: • Networks without layer normalization (LN) fail to train effectively, and ELU significantly outperforms ReLU for encoder learning. • In contrast, all dynamic training regimes only improve weaker architectures while failing to overcome fundamental architectural limitations.

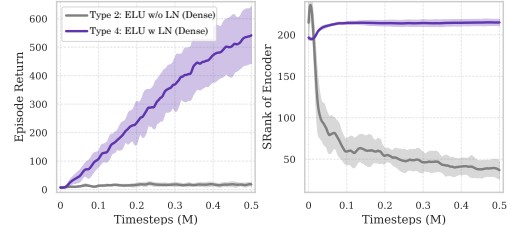

To understand why architectures without LN fail completely, we examine the representational capacity of the zsa network in the encoder module using the Stable-Rank (*SRank*) (Kumar et al., 2021), which measures the effective dimensionality of learned features. As shown in Figure 2, Type 2 networks suffer catastrophic representational collapse with rapidly declining *SRank*, while simply adding layer normalization (Type 4) prevents this collapse entirely. This fundamental difference reveals that preventing representational collapse in self-supervised RL depends not only on algorithmic techniques

Figure 2: Representational collapse causes ineffective training without layer normalization.

such as stop-gradient operations (Ni et al., 2024) and EMA-updated target encoders (Schwarzer et al., 2021), but also critically on appropriate DL mechanisms. Specifically, architectural improvements yield substantially greater benefits than introducing dynamic sparse training regimes.

**Scaling Behavior.** Based on base scale findings, we examine representative architectures (Types 2, 4, 6) across three orders of magnitude. As parameter counts increase, residual connections become essential for large encoder scalability. For Type 4 networks, dynamic sparse training regimes improve scalability, though these benefits stem primarily from network sparsity. Most notably, Type 6 networks maintain consistent scalability even at the largest scale with basic dense training alone. This progression demonstrates that architectural advances create inherently better optimization properties, enabling effective encoder scaling that dynamic sparse training strategies alone cannot achieve.

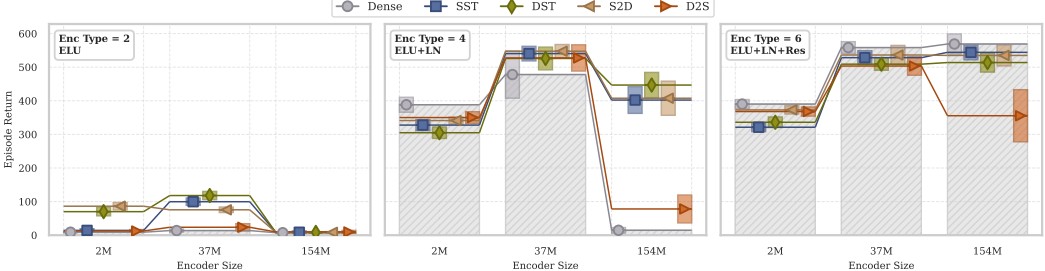

Figure 3: Scaling behavior of agent performance as encoder parameter count increases across three orders of magnitude. While introducing network sparsity and dynamicity can sometimes outperform dense training, architectural design remains the fundamental determinant of scalability potential.

> **Takeaway:** For encoders trained through self-supervised dynamics, architecture design inherently determines both representation quality and scalability, surpassing all dynamic training benefits.

## 3.2 CRITIC: DYNAMIC SPARSE TRAINING ENHANCES TD-BASED VALUE LEARNING

Critics trained through TD learning suffer unique pathologies due to bootstrapping from non-stationary targets (Lyle et al., 2023), motivating our investigation of how dynamic training approaches mitigate these specific issues. Based on Section 3.1 findings, we employ a stronger encoder network (Type 6, 37M parameters) for critic module investigation while maintaining default actor configurations.

**Comparison at Base Scale.** Unlike encoders where architectural differences prove dramatic, Figure 18 shows that critic networks achieve more similar performance across architectures at default

scale when supported by strong encoder representations. Training regime differences also remain minimal at this scale. This finding suggests that powerful encoder representations effectively capture approximate linear relationships between state-action pairs and their values, significantly reducing value learning complexity (Fujimoto et al., 2023; Scannell et al., 2024; Zheng et al., 2023).

**Scaling Behavior.** As critic networks scale up, Figure 4 reveals pronounced performance differences between network architectures and training strategies. Dense networks suffer severe degradation across most architectures, with catastrophic collapse in networks without residual connections (Types 1-4). Even advanced architectural combinations cannot overcome fundamental scaling barriers in dense training. In contrast, training strategies introducing network sparsity and dynamicity substantially enhance critic scalability across architectures. Among these approaches, ◆ DST demonstrates the most consistent improvements by maintaining constant sparsity while enabling dynamic topology adaptation, whereas ▶ D2S and ◀ S2D alter sparsity levels during training. These findings reveal a key insight about TD-based value learning: neither architectural innovations nor training regimes alone can overcome the critic scaling barrier. Unlocking full scaling potential of the critic module demands the right combination of both architectural design and training strategy.

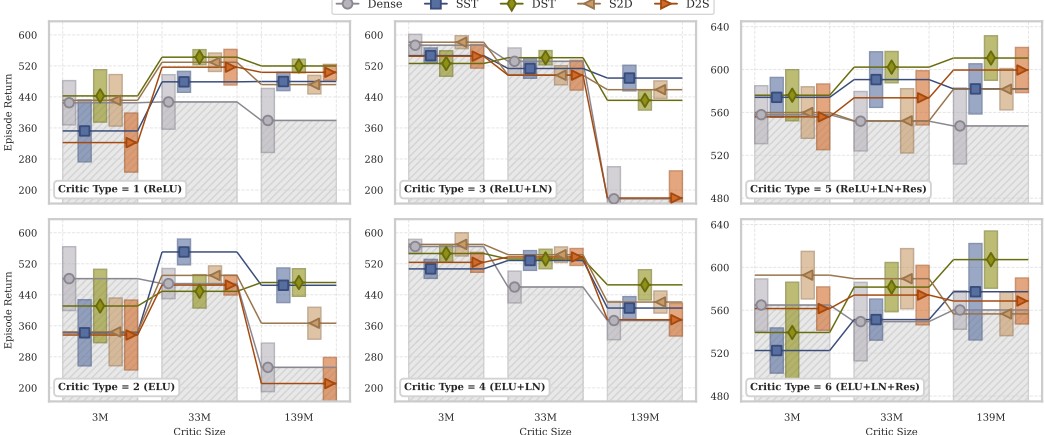

Figure 4: Dynamic training approaches, particularly DST, consistently outperform dense and static sparse baselines. Results demonstrate that unlocking the full potential of scaled critic networks requires the combination of advanced architectures with appropriate dynamic training regimes.

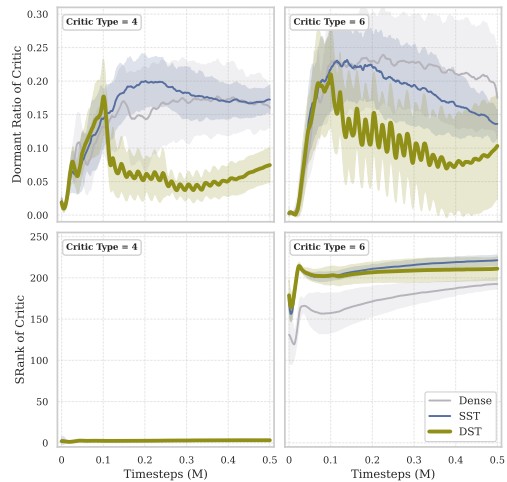

Figure 5: Optimization pathologies for the scaled critic networks with 139M parameters (scale=5).

To further understand the mechanisms behind these scaling behaviors, we examine critic pathologies using two key metrics: *SRank* (Kumar et al., 2021) for representational capacity and *Dormant Ratio* (Sokar et al., 2023) for plasticity loss. Figure 5 demonstrates that dynamic training effectively mitigates plasticity loss in large-scale critics, aligning with previous findings on pruning's benefits for value-based RL scalability (Ceron et al., 2024a). Meanwhile, network sparsity unlocks greater representational capacity, but only when combined with residual connections. Without these architectural safeguards, critics suffer catastrophic representational collapse even without severe plasticity loss. This dual mechanism explains why DST, combining network sparsity with dynamic topology adaptation, achieves superior scalability of the critic module.

> **Takeaway:** DST can notably enhance the scalability of TD-learned critics by mitigating plasticity loss while maintaining representational capacity. However, full scalability of critics requires the combination of appropriate dynamic sparse training with powerful architectural foundations.

### 3.3 ACTOR: STATIC SPARSITY OUTPERFORMS DYNAMIC TRAINING IN POLICY SCALING

Unlike critics and encoders, actor networks directly optimize deterministic policy gradients, creating distinct stability requirements (Touati et al., 2020). While dynamic training strategies proved beneficial for critic scaling, we investigate whether these same approaches effectively scale policy networks. For actor evaluation, we maintain a scaled Type 6 encoder (37M) and default Type 4 critic (3M) to isolate actor-specific effects while ensuring stable representation and value learning.

**Comparison at Base Scale.** As shown in Figure 22, LN consistently benefits actor networks across all training regimes. Interestingly, switching from ReLU to ELU activation functions significantly degrades actor performance, contrasting sharply with our findings for encoders and critics where ELU proved beneficial. This suggests that the feature sparsity induced by ReLU better supports policy gradient optimization, despite potential concerns about dormant neurons. In addition, training strategy differences remain minimal at the default scale across all architectural variants.

**Scaling Behavior.** Figure 6 reveals distinct scaling patterns for actor networks. First, normalization and residual connections remain essential for achieving scalable actors. However, unlike critics where dynamic training worked best, SST consistently outperforms all dynamic approaches for scaled actor networks. This advantage becomes more pronounced at larger scales, where the three dynamic training approaches exhibit more severe training instability.

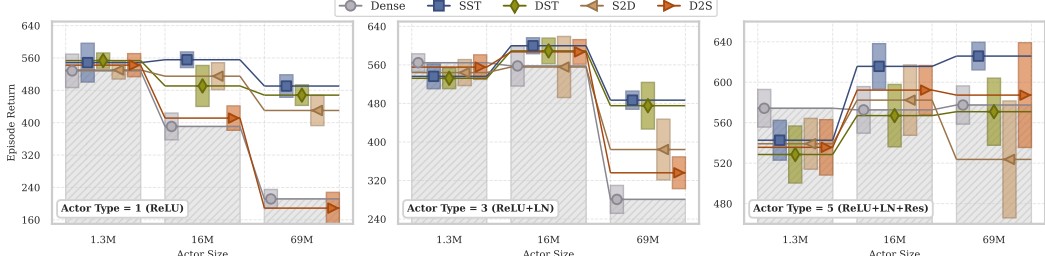

Figure 6: Actor network scaling performance across architectures and training regimes. Normalization and residual connections are critical for preventing collapse in large-scale actor networks. With this architectural foundation, SST consistently enables further effective scaling for policy networks.

To elucidate why SST outperforms dynamic approaches, Figure 7 examines plasticity loss at the largest scale. Type 3 networks without residual connections exhibit substantial dormant neuron proportions under dense training, while adding residual connections (Type 5) largely prevents this phenomenon, confirming that actors experience less severe plasticity issues than critics. Crucially, SST achieves the most consistent dormant neuron reduction across architectures while preserving training stability. Dynamic

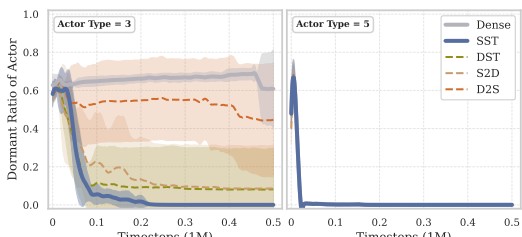

Figure 7: Network sparsity reduces dormant ratio and mitigates plasticity loss in scaled actors.

training approaches, while effective for critic plasticity, yield minimal benefits for actors and introduce detrimental training instability that disrupts policy optimization. This phenomenon explains SST's superiority and aligns with prior work (Ceron et al., 2024a) demonstrating that pruned networks effective in value-based RL fail to transfer successfully to actor-critic architectures.

> **Takeaway:** SST with well-designed architecture most effectively prevents actors from developing optimization pathologies during scaling by providing an ideal balance between reduced plasticity loss and stable gradient pathways essential for policy optimization.

### 3.4 INVESTIGATION REMARK

This investigation reframes our understanding of how dynamic sparse training impacts scalable DRL. Consistent with prior work (Ceron et al., 2024a; Liu et al., 2024), DST can significantly prevent basic MLP networks from developing severe optimization pathologies during scaling. However, these benefits are substantially overshadowed by architectural improvements alone. This demonstrates

that ***network architecture establishes the fundamental optimization dynamics most crucial for stable scaling***, aligning with recent scalable DRL advances that primarily stem from architectural innovations such as BRO (Nauman et al., 2024b) and Simba series (Lee et al., 2024; 2025).

Our second key contribution to current understanding is that while architecture is primary, architectural improvements alone can not suffice for maximizing DRL scalability. Instead, ***targeted training strategies with network sparsity and dynamicity can extract further scaling potential from strong architectural foundations***. Achieving this requires ***module-specific training strategies*** that align with each component's distinct learning paradigms and optimization characteristics.

## 4 METHOD: MODULE-SPECIFIC TRAINING FOR SCALABLE DEEP RL

Our investigation in Section 3 provides clear guidance for achieving scalable DRL through tailoring DL mechanisms without modifying underlying RL algorithms. This guidance operates on two essential levels: • **First,** establishing strong architectural foundations represents the fundamental prerequisite. As demonstrated throughout our investigation and consistent with prior studies (Lee et al., 2024; 2025; Nauman et al., 2024b; Wang et al., 2025), these elements establish the optimization dynamics that are most crucial for stable scaling. • **Second, and equally important,** we propose **Module-Specific Training (MST)** as a systematic framework that assigns targeted training strategies based on the distinct learning paradigms of each module. Specifically, MST applies dense training for the encoder, DST for the critic to address the most severe TD learning pathologies, and SST for the actor to balance the plasticity and stability during policy optimization.

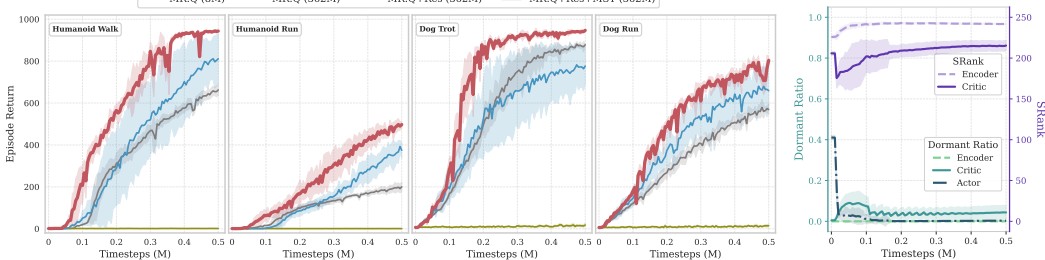

Figure 8: Scaling performance with massive model size (5× width and depth, 362M parameters). Baseline MR.Q fails completely at this scale, while adding residual connections effectively prevents catastrophic collapse. However, MST proves essential for unlocking substantial additional scalability gains on top of strong architectural foundations. The right panel shows MST maintains healthy optimization dynamics with high SRank and minimal dormant ratios even at this massive scale.

To validate the effectiveness and necessity of MST, we first scale MR.Q to massive size, as shown in Figure 8. Naively scaling up DRL models without tailored architecture and training strategies results in catastrophic failure, unlike the successful scaling observed in language and vision domains. These findings reveal that scalable

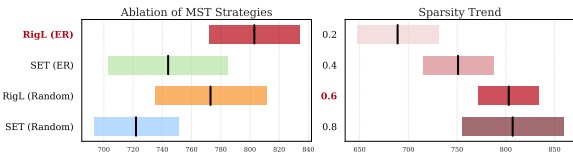

Figure 9: Ablation study on MST implementation details in MR.Q+Res+MST (362M) on Dog Run.

DRL demands both strong architectural foundations and strategic MST approaches. We conduct ablations for MST design choices, as shown in Figure 9. ER initialization outperforms random initial-

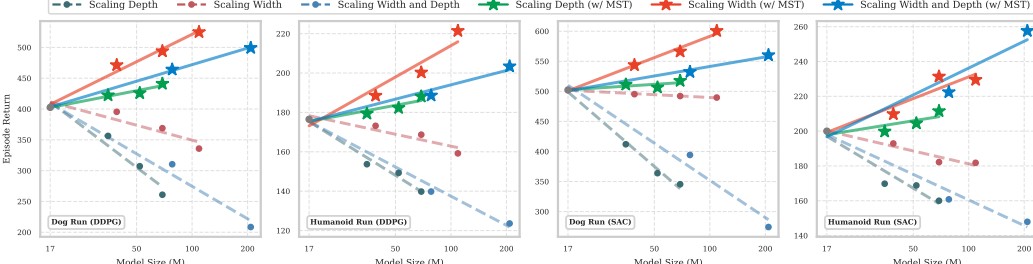

Figure 10: MST generalizes to DRL algorithms without decoupled encoders. On SAC and DDPG with Simba architectures, standard scaling approaches (dashed lines) lead to performance degradation, while MST (solid lines) enables successful scaling across different dimensions and tasks.

ization, and RigL (Evci et al., 2020) proves superior to Sparse Evolutionary Training (SET) (Mocanu et al., 2018) for the critic due to gradient-based growth. Consistent with Ma et al. (2025), increasing sparsity benefits massive DRL models while excessive sparsity potentially causes instability.

To demonstrate MST's broader applicability beyond algorithms with decoupled encoders, we evaluate its effectiveness on the widely-used SAC (Haarnoja et al., 2018) and DDPG (Lillicrap, 2015). Using Simba (Lee et al., 2024) as our strong architectural foundation and following sparsity configurations from Ma et al. (2025), we examine scaling trends across width, depth, and both dimensions simultaneously.

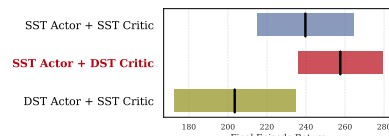

Figure 11: Validation of MST necessity on Humanoid Run (SAC, 207M).

As shown in Figure 10, naive scaling consistently degrades performance, while MST achieves stable scaling gains. In addition, Figure 11 validates the importance of targeted training strategy assignment: introducing dynamicity benefits the critic network while proving detrimental to the actor network. We also validate MST's effectiveness in MR.SAC, which combines model-based representation learning from MR.Q with the SAC algorithm, with detailed results provided in Appendix E.

## 5 CONCLUSION AND DISCUSSION

Dynamic sparse training (DST) was initially developed to reduce training costs with minimal performance degradation. However, DRL researchers discovered sparse networks surprisingly outperformed dense counterparts (Graesser et al., 2022; Tan et al., 2023). Recent studies further revealed that such dynamic training strategies can alleviate the scaling barriers caused by DRL's unique optimization pathologies (Ceron et al., 2024a; Liu et al., 2024). Concurrently, scalable DRL has achieved significant progress, particularly through architectural advances. Against this backdrop, a critical question arises: *what role does DST play now?* Our investigation addresses this question and reveals two key insights. First, DST alone cannot match architectural improvements; instead, it serves as a complementary mechanism that unlocks additional potential. Second, effective dynamic training demands module-specific approaches: different RL modules require distinct training strategies tailored to their unique learning paradigms. We distill these insights into an empirical MST framework and validate its effectiveness in improving scalability across multiple base RL algorithms.

**Lessons.** Beyond technical implementation details, our investigation further reveals broader principles that fundamentally reshape how we understand and implement RL-tailored DL mechanisms.

● *Lesson 1: Unlocking the full potential of DRL requires carefully designed DL mechanisms tailored to the unique challenges of RL optimization landscapes.* Our experiments show that standard neural architectures developed for supervised learning fail to accommodate the non-stationarity and bootstrapping inherent to RL. Specialized elements like layer normalization and tailored training strategy deliver improvements beyond what algorithmic refinements alone can achieve. This confirms that DRL optimization pathologies require distinct solutions beyond conventional DL practices. Beyond our focus on architecture and training regimes, other DL components such as optimizers (Ellis et al., 2024; Goldie et al., 2024; Muppidi et al., 2024; Castanyer et al., 2025) and regularization techniques (Chung et al., 2024) also represent promising solutions.

● *Lesson 2: Different learning paradigms exhibit distinct optimization characteristics that demand module-specific architectural and training considerations.* Despite operating within a unified system, different DRL modules follow distinct learning objectives that create fundamentally different optimization dynamics. For instance, TD-learned critics suffer more severely from plasticity loss than other modules, while actors require stable gradient pathways for effective policy learning. Hence, effective RL-tailored mechanisms must recognize that different components within the same agent require distinct architectural and training approaches.

● *Lesson 3: With basic optimization pathologies in standard DRL tasks now effectively addressed, it is time to advance toward more challenging real-world scenarios.* Our results in Section 4 demonstrate that combining advanced architectures with MST enables massive model scaling while maintaining healthy optimization dynamics. However, real-world applications present fundamentally greater challenges such as larger action spaces (Farquhar et al., 2020), multi-task conflicts (Joshi et al., 2025) and continual adaptation requirements (Elsayed & Mahmood, 2024; Abel et al., 2023). Therefore, future research should focus on developing more advanced RL-tailored DL mechanisms that preserve the benefits of scaling under these demanding optimization challenges.

## ETHICS STATEMENT

All authors have read and adhered to the ICLR Code of Ethics. Our research does not involve human subjects, personally identifiable information, or sensitive personal data.

## REPRODUCIBILITY STATEMENT

We provide comprehensive implementation details and experimental configurations in the appendix, including network architectures, hyperparameters, and training procedures. All experiments were conducted with multiple random seeds to ensure statistical reliability. Source code for reproducing our experimental results will be made publicly available upon acceptance.

## THE USE OF LARGE LANGUAGE MODELS

Large language models were used during manuscript preparation for language editing, including grammar correction, sentence restructuring, and clarity improvements. All technical content, experimental design, analysis, and conclusions are the original work of the authors.

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

The appendix is divided into several sections, each giving extra information and details.

## A   RELATED WORK AND BACKGROUND

This section examines the key research areas that inform our work. We first explore the distinctive optimization pathologies that emerge specifically in deep reinforcement learning (DRL) environments. We then analyze the fundamental scaling challenges that have limited DRL model expansion and survey recent innovations aimed at overcoming these barriers. Finally, we review existing applications of dynamic sparse training in DRL, whether aimed at model compression, training acceleration, or performance enhancement.

### A.1   OPTIMIZATION PATHOLOGIES IN DRL

Although deep neural networks have driven remarkable advances in current deep reinforcement learning (DRL) applications, growing evidence indicates that DRL networks are prone to severe optimization pathologies during training (Nikishin, 2024; Nauman et al., 2024a; Goldie et al., 2024). These pathologies emerge from the unique challenges of integrating DL mechanisms with the RL paradigm, presenting distinctive issues not encountered in traditional RL settings. Such difficulties stem from fundamental characteristics of RL that distinguish it from supervised learning: non-stationary data distributions and optimization objectives, as well as the inherent nature of learning through online interactions.

These DRL-specific pathologies have recently been identified and characterized through various phenomena: primacy bias (Nikishin et al., 2022; Yuan et al., 2025), the dormant neuron phenomenon (Sokar et al., 2023), implicit under-parameterization (Kumar et al., 2021), capacity loss (Lyle et al., 2022), and the broader issue of plasticity loss (Klein et al., 2024; Abbas et al., 2023; Juliani & Ash, 2024). Although these studies approach the problem from different angles, they converge on a common finding: DRL networks routinely develop severe optimization pathologies during training that fundamentally impair their ability to learn from new experiences. The consequences manifest either as severe sample inefficiency or, in the worst cases, complete learning stagnation (Ma et al., 2024). These pathologies manifest through several observable symptoms: a high proportion of

inactive neurons (Sokar et al., 2023), reduced effective rank of representational features (Kumar et al., 2021; Lyle et al., 2022), unbounded growth in parameter norms (Lyle et al., 2023), and increased gradient interference across training samples (Lyle et al., 2024b). Each of these symptoms contributes to the network's diminishing ability to effectively learn and optimize both the policy and value functions.

## A.2 UNIQUE SCALING BARRIER IN DRL

Using deep neural networks is a key factor in successfully applying reinforcement learning to complex tasks. However, while some recent advances in supervised learning have been driven by scaling up the number of network parameters, a phenomenon commonly referred to as *scaling laws*, it remains challenging to increase the number of parameters in deep reinforcement learning without experiencing performance degradation. Several recent works in DRL have addressed this by scaling up network sizes through various strategies. Schwarzer et al. (2023) transitioned from the original CNN architecture to the ResNet-based Impala-CNN architecture (Espeholt et al., 2018) and scaled the network width by a factor of 4. Both BRO (Nauman et al., 2024b) and SimBa (Lee et al., 2024) employed deeper networks that incorporate layer normalization (Lei Ba et al., 2016) and residual connections. Ceron et al. (2024b) incorporated a soft Mixture-of-Experts module (Puigcerver et al., 2020) into value-based networks, resulting in more parameter-scalable models and improved performance. Farebrother et al. (2024) show that value functions trained using categorical cross-entropy substantially enhance performance and scalability in multiple domains. Ceron et al. (2024a) utilized magnitude pruning on value-based networks, progressively decreasing the number of parameters in dense architectures during training to achieve highly sparse models, leading to improved performance when scaling network width.

## A.3 DYNAMIC SPARSE TRAINING IN DRL

Initial explorations of sparse networks in DRL were primarily motivated by the potential for model compression, aiming to accelerate training and facilitate efficient model deployment (Tan et al., 2023). Early explorations of network sparsification in DRL primarily focused on behavior cloning and offline RL settings (Arnob et al., 2021; Vischer et al., 2021). In the more challenging context of online RL, Sokar et al. (2021) explored the application of Sparse Evolutionary Training (SET) (Mocanu et al., 2018) and successfully achieved 50% sparsity. However, attempts to increase sparsity beyond this level resulted in significant training instability. Subsequently, Tan et al. (2023) enhanced the efficacy of dynamic sparse training through a novel delayed multi-step temporal difference target mechanism and a dynamic-capacity replay buffer, ultimately achieving sparsity levels of up to 95%. Graesser et al. (2022) conducted a comprehensive investigation and demonstrated that pruning consistently outperforms standard dynamic sparse training methods, such as SET (Mocanu et al., 2018) and RigL (Evci et al., 2020). Data Adaptive Pathway Discovery (DAPD) (Arnob et al., 2024) dynamically adjusts network pathways in response to online RL distribution shifts, maintaining effectiveness at high sparsity levels.

Beyond the initial goal of achieving parameter-efficient architectures through sparsity, recent studies have recognized that sparse and adaptive networks can enhance DRL model scalability while mitigating training pathologies such as plasticity loss. For instance, Ceron et al. (2024a) shows that applying gradual magnitude pruning to large models significantly enhances the performance of value-based agents. Similarly, Ceron et al. (2024b) demonstrates that incorporating Soft MoEs into value-based RL networks enables better parameter scaling. Furthermore, Neuroplastic Expansion (Liu et al., 2024) addresses plasticity challenges by progressively evolving networks from sparse to dense architectures, effectively leveraging increased model capacity. Although these approaches differ in implementation, they all fall under the broader category of dynamic sparse training, where network topology evolves during training.

# B IMPLEMENTATION DETAILS FOR TRAINING METHODS AND NETWORK ARCHITECTURES

In this section, we provide comprehensive implementation details essential for reproducing our experimental results and understanding the technical foundations of our research. We first present the

implementation specifics of both dynamic and static training regimes, including sparsity mechanisms, pruning schedules, and topology adaptation procedures (Section B.1). Next, we detail the architectural specifications for our network variants, analyzing the design considerations behind each component and their interactions (Section B.2). Finally, we describe the implementation and hyperparameter configurations of our base algorithm MR.Q (Fujimoto et al., 2025), covering the core mechanism that enables module-specific training paradigms in reinforcement learning (Section B.3). These details collectively establish the technical framework upon which our investigation into dynamic training regimes for reinforcement learning is built.

## B.1 DYNAMIC AND STATIC TRAINING IMPLEMENTATION DETAILS

### B.1.1 STATIC SPARSE TRAINING (SST) WITH ONE-SHOT RANDOM PRUNING

Static sparse training with one-shot random pruning creates binary masks for each layer at initialization. These masks define the network's sparse topology and remain unchanged throughout training. For a network with $L$ layers, each layer $l$ has a binary mask $\mathbf{M}^l \in \{0,1\}^{n^l \times n^{l-1}}$, where $n^l$ represents the number of units in layer $l$. The effective weights during both training and inference are calculated as $\mathbf{W}^l_{\text{eff}} = \mathbf{M}^l \odot \mathbf{W}^l$, where $\odot$ denotes element-wise multiplication.

Random pruning implements a straightforward sampling process that requires only predetermined layer-wise sparsity ratios. Two common approaches for determining these ratios from the overall network sparsity are:

1. **Uniform**: Each layer $l$ uses the same sparsity ratio $s^l$ equal to the overall network sparsity $S$.

2. **Erdős-Rényi (ER)**: This method generates sparse masks where the sparsity in each layer $s^l$ is proportional to $1 - \frac{n^{l-1}+n^l}{n^{l-1}n^l}$ for fully-connected layers (Mocanu et al., 2018) and to $1 - \frac{n^{l-1}+n^l+w^l+h^l}{n^{l-1}n^l w^l h^l}$ for convolutional layers with kernel dimensions $w^l \times h^l$ (Evci et al., 2020).

Multiple studies in both supervised learning (Liu et al., 2022) and reinforcement learning (Tan et al., 2023; Liu et al., 2024) have demonstrated that ER-based initialization consistently outperforms uniform sparsity, particularly at high sparsity levels. Therefore, we adopt ER-based layer-wise sparsity ratios throughout our experiments, with a fixed overall sparsity level of 0.6, which our preliminary experiments identified as an effective balance between network size and performance.

### B.1.2 IMPLEMENTATION OF RIGL (EVCI ET AL., 2020) FOR DYNAMIC SPARSE TRAINING

Similar to SST, DST begins with a randomly pruned network using ER. However, while SST maintains a fixed topology throughout training, DST dynamically evolves topology while preserving overall sparsity.

---

**Algorithm 1** Dynamic Sparse Training based on RigL (Evci et al., 2020)

---

1: $N_l$: Number of parameters in layer $l$
2: $\theta_l$: Parameters in layer $l$
3: $M_{\theta_l}$: Sparse mask of layer $l$
4: $s_l$: Sparsity of layer $l$
5: $L$: Loss function
6: $\zeta_t$: Update fraction in training step $t$
7: **for** each layer $l$ **do**
8: $\quad k = \zeta_t(1 - s_l)N_l$
9: $\quad \mathbb{I}_{\text{drop}} = \text{ArgTopK}(-|\theta_l \odot M_{\theta_l}|, k)$
10: $\quad \mathbb{I}_{\text{grow}} = \text{ArgTopK}_{i \notin \theta_l \odot M_{\theta_l} \setminus \mathbb{I}_{\text{drop}}}(|\nabla_{\theta_l}L, k|)$
11: $\quad$ Update $M_{\theta_l}$ according to $\mathbb{I}_{\text{drop}}$ and $\mathbb{I}_{\text{grow}}$
12: $\quad \theta_l \leftarrow \theta_l \odot M_{\theta_l}$
13: **end for**

---

The core mechanism of RigL (Evci et al., 2020) involves periodically updating the network topology during training. Every $N$ timesteps, a fraction of connections are modified through a coordinated drop-and-grow process: the lowest magnitude weights are pruned, and an equal number of previously inactive connections with the highest gradient magnitudes are activated (Dettmers & Zettlemoyer, 2019). This allows the network to reallocate parameters toward more important connections discovered during the learning process. The fraction of weights updated at each interval (drop fraction $\zeta_t$) follows a cosine decay schedule throughout training:

$$\zeta_t = \zeta_{\text{final}} + \frac{1}{2}(\zeta_{\text{initial}} - \zeta_{\text{final}})(1 + \cos(\pi \cdot t/T)) \tag{1}$$

where $t$ is the current step and $T$ is the total number of training steps. This schedule enables greater exploration early in training and more exploitation later, helping the network converge to a stable topology. Algorithm 1 presents this process in detail. To ensure consistent comparison across different network architectures and training regimes, we maintain the same overall sparsity level (0.6) throughout our experiments. This level provides a balanced trade-off between parameter efficiency and network expressivity, ensuring that sparsity itself doesn't become a limiting factor in performance.

### B.1.3 Extending RigL Framework to Dense-to-Sparse and Sparse-to-Dense

While Dynamic Sparse Training (DST) maintains constant sparsity, we extend the RigL framework to implement two additional dynamic training regimes with changing sparsity levels: Dense-to-Sparse (D2S) and Sparse-to-Dense (S2D) training. Both approaches follow the same core drop-and-grow mechanism as RigL but incorporate a time-dependent sparsity schedule (Ceron et al., 2024a). The sparsity at training step $t$ is calculated according to:

$$s_t = s_f + (s_i - s_f)\left(1 - \frac{t - t_{start}}{t_{end} - t_{start}}\right)^{\lambda} \tag{2}$$

where $s_i$ is the initial sparsity, $s_f$ is the final sparsity, $t_{start}$ and $t_{end}$ define the scheduling period, and $\lambda$ controls the non-linearity of the transition.

For Dense-to-Sparse (D2S) training, we set $s_i = 0$ and $s_f = 0.6$, starting with a fully dense network and gradually increasing sparsity. This approach allows the network to establish important connections early in training before progressively removing less useful parameters. The pruning process follows magnitude-based criteria similar to RigL but adjusts the number of pruned connections at each update step to match the target sparsity schedule. Conversely, Sparse-to-Dense (S2D) training sets $s_i = 0.6$ and $s_f = 0$, beginning with a sparse network and gradually increasing connectivity throughout training. This approach follows the neuroplastic expansion hypothesis that releasing additional capacity during learning can help networks escape optimization pathologies while maintaining training efficiency. The growth process continues to use gradient information to identify promising connections, but more connections are added than removed at each update to follow the decreasing sparsity schedule. Both approaches maintain the overall RigL topology update mechanism while adapting the number of parameters added or removed at each update step. For our experiments, we set $\lambda = 2$.

### B.2 Network Architecture Specifications and Analysis

This section discusses key architectural components that significantly impact deep reinforcement learning performance, focusing on activation functions and residual connections that help mitigate optimization pathologies.

### B.2.1 Activation Function: ReLU vs. ELU

Activation functions play a critical role in deep neural networks by introducing non-linearity and regulating information flow. In deep reinforcement learning, where optimization pathologies are particularly severe, the choice of activation function can significantly influence network plasticity and learning dynamics.

The Rectified Linear Unit (ReLU) has been the dominant activation function due to its computational efficiency and effectiveness in addressing vanishing gradient problems:

$$\text{ReLU}(x) = \max(0, x) \quad (3)$$

However, ReLU suffers from a fundamental limitation: it produces zero gradients for all negative inputs, leading to the "dormant neuron" problem where neurons become permanently inactive during training (Sokar et al., 2023). This issue is particularly problematic in DRL, where non-stationary targets can drive many neurons into inactive states.

The Exponential Linear Unit (ELU) addresses this limitation by providing non-zero outputs and gradients for negative inputs:

$$\text{ELU}(x) = \begin{cases} x & \text{if } x > 0 \\ \alpha(e^x - 1) & \text{if } x \leq 0 \end{cases} \quad (4)$$

where $\alpha$ is typically set to 1. The negative saturation in ELU ensures that inactive neurons can be "revived" during training as gradients can still flow through them, making the network more robust to shifting data distributions in reinforcement learning settings.

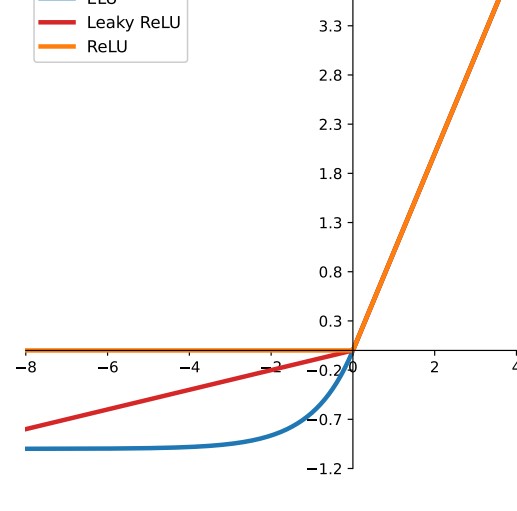

Figure 12: Comparison of activation functions: ReLU (red) produces zero output and gradients for negative inputs, while ELU (blue) provides smooth, non-zero negative outputs bounded at $-\alpha$ (typically $-1$). This difference is critical for preventing dormant neurons in reinforcement learning.

Our experiments show that ELU consistently benefits encoder and critic by maintaining higher representational capacity, while surprisingly, ReLU remains more effective for actor networks by providing beneficial sparsity for policy optimization (see Section 3.3).

### B.2.2 Pre-Layer Residual Connection vs. Post-Layer Residual Connection

Residual connections are crucial for mitigating gradient flow issues in deep networks, but their implementation details can significantly impact learning dynamics in reinforcement learning. We examine two common approaches to residual connections that differ in their placement within network blocks.

In Pre-Layer Residual Connection (Figure 13 a), the identity shortcut bypasses normalization and activation functions, similar to the approach used in Simba (Lee et al., 2024). The forward pass can be expressed as:

$$y = F(x) + x \quad (5)$$

where $F(x)$ represents the entire transformation sequence (normalization, activation, and linear layers). This approach preserves the raw input information through the residual pathway, allowing direct access to unmodified representations.

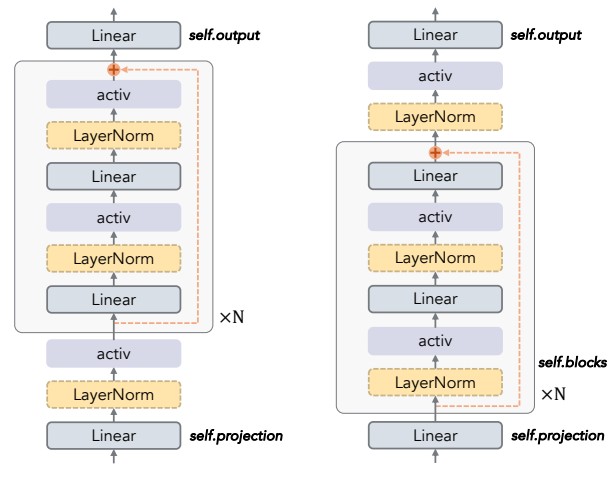

(a) Pre-Layer Residual Connection    (b) Post-Layer Residual Connection

Figure 13: Architectural comparison of (a) Pre-Layer Residual Connection, where identity shortcuts bypass normalization and activation, and (b) Post-Layer Residual Connection, where normalization and activation are applied before the residual pathway. The key difference lies in whether raw input information is preserved in the residual path.

In contrast, Post-Layer Residual Connection (Figure 13 b) applies normalization and activation before adding the identity shortcut, similar to the implementation in BRO (Nauman et al., 2024b). The computation flow can be described as:

$$y = G(F'(x)) + F'(x) \tag{6}$$

where $F'(x)$ represents the initial transformation (typically normalization and activation), and $G$ represents subsequent linear layers. This approach normalizes inputs before the residual pathway, potentially stabilizing the representation scale throughout the network.

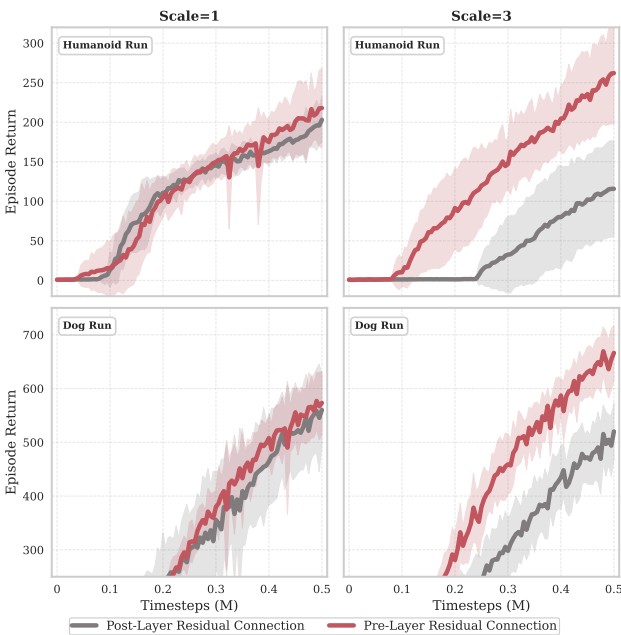

Figure 14: Performance comparison between pre-layer and post-layer residual connections on Humanoid Run and Dog Run tasks at default scale and larger scale. pre-layer connections consistently deliver better performance, with advantages becoming more pronounced as model size increases.

Our experimental results (Figure 14) demonstrate that Pre-Layer Residual Connection consistently outperforms Post-Layer implementation across tasks, particularly when scaling network depth. The performance gap widens significantly at larger scales (3x), where unimpeded gradient flow becomes increasingly important. Pre-Layer connections maintain stronger learning signals through deep networks by avoiding potential representation bottlenecks that can occur when normalization layers precede the residual pathway.

## B.3 Implementation and Hyperparameters of Base Algorithm MR.Q

In order to achieve a general-purpose model-free reinforcement learning algorithm, MR.Q (Model-based Representations for Q-learning) (Fujimoto et al., 2025) was proposed with decoupled model-based representation learning. It learns a state encoder $f_\omega(s)$ and a state-action encoder $g_\omega(z_s, a)$ to construct unified latent state embedding $z_s$ and state-action embedding $z_{sa}$, respectively. Following these encoders, decoupled reinforcement learning components including value $Q_\theta(z_{sa})$ and policy $\pi_\phi(z_s)$ are learned in a TD7 manner. The encoders are trained with self-predictive dynamics learning refering from model-based RL. Specifically, at timestep $t-1$, given output state-action embedding $\tilde{z_{sa}}^{t-1} = g_\omega(f_\omega(s^{t-1}), a^{t-1})$, MR.Q try to predict the state embedding $\tilde{z_s}^t$, reward $\tilde{r}^t$, terminal $\tilde{d}^t$ at the next timestep $t$ with a predictor parameterised by $\mathbf{m}$:

$$\tilde{z_s}^t, \tilde{r}^t, \tilde{d}^t = g_\omega(\tilde{z}^{t-1}, a^{t-1})^T \mathbf{m}. \tag{7}$$

And the encoder loss is calculated by a weighted-sum losses of each item along horizon $H_{enc}$:

$$L_{enc}(f_\omega, g_\omega, \mathbf{m}) = \sum_{t=1}^{H_{env}} \lambda_{rwd} L_{rwd}(\tilde{r}^t) + \lambda_{dnm} L_{dnm}(\tilde{z}_s^t) + \lambda_{tmn} L_{tmn}(\tilde{d}^t). \tag{8}$$

Here reward loss adopts the cross entropy between the predicted reward and a two-hot encoding of the true reward:

$$L_{rwd}(\tilde{r}) = \text{CE}(\tilde{r}, \text{Two-Hot}(r)). \tag{9}$$

Dynamics loss and terminal loss are both mean squared error (MSE) over prediction and ground truth:

$$
\begin{aligned}
L_{dnm}(\tilde{z}_s) &= (\tilde{z}_s - z_s)^2, \\
L_{tmn}(\tilde{d}) &= (\tilde{d} - d)^2.
\end{aligned}
\tag{10}
$$

Although with learned dynamics, as a model-free RL algorithm, MR.Q doesn't perform any planning or rollout simulated trajectories, which avoids introducing extra substantial computation as in model-based methods. Instead, it only utilise reward and dynamics losses as auxiliary tasks to learn the intermediate representation space. The model-based representation learning manners can not only provide richer learning signals, but also help decouple state embedding from original state spaces which might be quite depending on characteristics of different environments, making it possible for MR.Q being an general algorithm working well with different tasks given one set of hyperparameters.

For value function, MR.Q adopts loss in TD3 with a few modification like multi-step returns over a horizon $H_Q$ and replacing MSE with huber loss:

$$L_{value}(\tilde{Q}_i) = \text{Huber}(\tilde{Q}_i, \frac{1}{\bar{r}}(\sum_{t=0}^{H_Q-1} \gamma^t r_t + \gamma^{H_Q} \tilde{Q}_j')), \quad \tilde{Q}_j' = \bar{r}' \min_{j=1,2} Q_{\theta_j'}(z_{s_{H_Q} a_{H_Q}}), \tag{11}$$

where $\bar{r}$ denotes the reward scale computed over recent expreiences in the replay buffer. It is used to normalise the temporal difference target in the value update, thus stablise the updating process. In addition, it also plays a role to unify value learning in environments with very different reward magnitudes. And $\bar{r}'$ denotes the target reward scale, which helps the target value being stable even the reward scale $\bar{r}$ changes dramatically. $\bar{r}'$ are periodically updated with target network.

Policy is optimised by deterministic policy gradient with additional regularisation penalty on pre-activations $z_\pi$:

$$L_{policy}(a_\pi) = -\frac{1}{2} \sum_{i=\{1,2\}} \tilde{Q}_i(z_{sa_\pi}) + \lambda_{reg} z_\pi^2. \tag{12}$$

The regularization is added to avoid local minima when the reward is sparse.

A comprehensive evaluation of MR.Q on four common RL benchmarks with different observation and action spaces with a fix set of hyperparameters show the compatitive performance of MR.Q over other state-of-the-art RL algorithms including TD7 (Fujimoto et al., 2023), DrearmerV3 (Hafner et al., 2023), and TD-MPC2 (Hansen et al., 2023), etc. Notably, Compared with its model-based counterpart, MR.Q has much fewer parameters and much faster training and evaluation speed. Table 1 presents the algorithm hyperparameters from the original MR.Q implementation that remain consistent across all our experiments.

Table 1: **Algorithm Hyperparameters for MR.Q.**

| | Hyperparameter | Value |
|---|---|---|
| | Dynamics loss weight $\lambda_{\text{Dynamics}}$ | 1 |
| | Reward loss weight $\lambda_{\text{Reward}}$ | 0.1 |
| | Terminal loss weight $\lambda_{\text{Terminal}}$ | 0.1 |
| | Pre-activation loss weight $\lambda_{\text{pre-activ}}$ | $1e-5$ |
| | Encoder horizon $H_{\text{Enc}}$ | 5 |
| | Multi-step returns horizon $H_Q$ | 3 |
| TD3 | Target policy noise $\sigma$ | $\mathcal{N}(0, 0.2^2)$ |
| | Target policy noise clipping $c$ | $(-0.3, 0.3)$ |
| LAP | Probability smoothing $\alpha$ | 0.4 |
| | Minimum priority | 1 |
| Exploration | Initial random exploration time steps | 10k |
| | Exploration noise | $\mathcal{N}(0, 0.2^2)$ |
| Common | Discount factor $\gamma$ | 0.99 |
| | Replay buffer capacity | 1M |
| | Mini-batch size | 256 |
| | Target update frequency $T_{\text{target}}$ | 250 |
| | Replay ratio | 1 |

## C SETUP

To implement the six network architecture variants described in Section B.2, we modified the MR.Q network architecture to flexibly incorporate different combinations of normalization, activation functions, and residual connections. We developed a modular implementation that allows us to systematically investigate the impact of each architectural element while maintaining the core functionality of the original algorithm.

First, we implement the `BaseMLPBlock` class, which serves as the fundamental building block in our architecture. This component encapsulates a configurable network block with options for layer normalization and activation functions, enabling systematic investigation of their impact on network performance.

```python
1  class BaseMLPBlock(nn.Module):
2      """
3      Base block for MLP networks with optional layer normalization
4      and activation functions
5      """
6      def __init__(self, hidden_size: int,
7                   use_layer_norm: bool = True,
8                   activation: str = 'elu'):
9          super().__init__()
10
11         self.use_layer_norm = use_layer_norm
12         self.activ = getattr(F, activation)
13
14         # Two linear layers with optional layer norm
15         self.linear1 = nn.Linear(hidden_size, hidden_size)
16         self.linear2 = nn.Linear(hidden_size, hidden_size)
17
18         self.apply(weight_init)
19
20     def forward(self, x: torch.Tensor):
21         # Apply optional LN and activation before first layer
22         if self.use_layer_norm:
23             y = ln_activ(x, self.activ)
24         else:
```

```
25              y = self.activ(x)
26
27          # First layer
28          if self.use_layer_norm:
29              y = ln_activ(self.linear1(y), self.activ)
30          else:
31              y = self.activ(self.linear1(y))
32
33          # Second layer (only linear transformation, without
34          # normalization or activation)
35          y = self.linear2(y)
36
37          return y
```

The `BlockMLP` module composes multiple `BaseMLPBlock` instances to create complete network architectures. It implements pre-layer residual connections and handles the projections between dimensions. This design allows us to scale network depth while maintaining consistent behavior across different architectural configurations.

```
1  class BlockMLP(nn.Module):
2      """
3      MLP with multiple residual blocks using
4      pre-layer residual connections
5      """
6      def __init__(self,
7                   input_dim: int,
8                   output_dim: int,
9                   hidden_dim: int,
10                  use_layer_norm: bool = True,
11                  activation: str = 'elu',
12                  num_blocks: int = 1,
13                  use_residual: bool = False):
14         super().__init__()
15
16         self.use_residual = use_residual
17         self.activ = getattr(F, activation)
18         self.use_layer_norm = use_layer_norm
19
20         # Projection layer - maps input to hidden dimension
21         self.projection = nn.Linear(input_dim, hidden_dim)
22
23         # Multiple residual blocks
24         self.blocks = nn.ModuleList()
25         for _ in range(num_blocks):
26             self.blocks.append(BaseMLPBlock(hidden_dim,
27                                             use_layer_norm,
28                                             activation))
29
30         # Output layer
31         self.output = nn.Linear(hidden_dim, output_dim)
32
33         self.apply(weight_init)
34
35     def forward(self, x: torch.Tensor):
36         # Initial projection
37         x = self.projection(x)
38
39         # Process through blocks with
40         # Pre-Layer Residual Connections
41         for block in self.blocks:
42             # Store input to the block for residual connection
43             block_input = x
44
45             # Get output of the block
46             # (includes internal LayerNorm + activ operations)
```

```
47            block_output = block(x)
48
49                # Apply residual connection
50                # right after linear transformations
51                if self.use_residual:
52                    x = block_output + block_input
53                else:
54                    x = block_output
55
56            # Apply LN and activation after all blocks are processed
57            if self.use_layer_norm:
58                x = ln_activ(x, self.activ)
59            else:
60                x = self.activ(x)
61
62            # Final output layer
63            return self.output(x)
```

Finally, we define the `ARCH_CONFIGS` dictionary that specifies the six architecture types corresponding to Figure 1. Each type represents a specific combination of activation function (ReLU vs. ELU), layer normalization (with vs. without), and residual connections (with vs. without), enabling systematic comparison of their effects on performance.

```
1  # Define network architecture configurations
2  ARCH_CONFIGS = {
3      1: {
4          "use_layer_norm": False,
5          "activation": "relu",
6          "use_residual": False
7      },  # Vanilla MLP (ReLU)
8
9      2: {
10          "use_layer_norm": False,
11          "activation": "elu",
12          "use_residual": False
13      },  # Vanilla MLP (ELU)
14
15      3: {
16          "use_layer_norm": True,
17          "activation": "relu",
18          "use_residual": False
19      },  # MLP with Layer Norm (ReLU)
20
21      4: {
22          "use_layer_norm": True,
23          "activation": "elu",
24          "use_residual": False
25      },  # MLP with Layer Norm (ELU)
26
27      5: {
28          "use_layer_norm": True,
29          "activation": "relu",
30          "use_residual": True
31      },  # MLP with Layer Norm and Residual Connections (ReLU)
32
33      6: {
34          "use_layer_norm": True,
35          "activation": "elu",
36          "use_residual": True
37      },  # MLP with Layer Norm and Residual Connections (ELU)
38 }
```

Table 2: **Network Hyperparameters for MR.Q.** The default configuration corresponds to Encoder Type 4 (scale=1), Critic Type 4 (scale=1), and Actor Type 3 (scale=1) in our experiments. Parameters in shaded rows (hidden dimensions and block numbers) are scaled by the corresponding factor (3× or 5×) in scalability experiments.

| | Hyperparameter | Value |
|---|---|---|
| | Optimizer | AdamW |
| | Learning rate | $1e-4$ |
| | Weight decay | $1e-4$ |
| | $\mathbf{z}_s$ dim | 512 |
| | $\mathbf{z}_s$ block num | 1 |
| | $\mathbf{z}_{sa}$ dim | 512 |
| | $\mathbf{z}_{sa}$ block num | 1 |
| Encoder Network | $\mathbf{z}_a$ dim (only used within architecture) | 256 |
| | Hidden dim | 512 |
| | Activation function | ELU |
| | Layer Normalization | True |
| | Residual Connection | False |
| | Weight initialization | Xavier uniform |
| | Bias initialization | 0 |
| | Reward bins | 65 |
| | Reward range | $[-10, 10]$ (effective: $[-22\text{k}, 22\text{k}]$) |
| | Optimizer | AdamW |
| | Learning rate | $3e-4$ |
| | Hidden dim | 512 |
| | Block num | 1 |
| Value Network | Activation function | ELU |
| | Layer Normalization | True |
| | Residual Connection | False |
| | Weight initialization | Xavier uniform |
| | Bias initialization | 0 |
| | Gradient clip norm | 20 |
| | Optimizer | AdamW |
| | Learning rate | $3e-4$ |
| | Hidden dim | 512 |
| | Block num | 1 |
| Policy Network | Activation function | ReLU |
| | Layer Normalization | True |
| | Residual Connection | False |
| | Weight initialization | Xavier uniform |
| | Bias initialization | 0 |
| | Gumbel-Softmax $\tau$ | 10 |

Table 2 details the network-specific hyperparameters. The default configuration shown corresponds to Encoder Type 4 (scale=1), Critic Type 4 (scale=1), and Actor Type 3 (scale=1) in our implementation. These parameters are systematically varied throughout our experiments to evaluate different architecture types and training regimes. For scalability experiments, we multiply the hidden dimensions and block numbers (highlighted in gray) by the corresponding scale factor (3× or 5×).

# D  DETAILED INVESTIGATION RESULTS

This section provides comprehensive visualization of experimental results across different network components, architectures, and training regimes. We present detailed performance data from Dog Run and Humanoid Run tasks that supports the main findings discussed in the paper. All experiments were conducted with 8 random seeds to ensure statistical reliability, with error bars representing one standard deviation.

## D.1  ENCODER

**Default Scale.**    As shown in Figure 15, the average performance comparison between Dog Run and Humanoid Run tasks reveals the striking dominance of architectural design over training regimes for encoder performance. Overall, non-saturating ELU activation function and layer normalization prove essential for developing powerful encoders, while residual connections offer minimal benefits at this scale. In contrast, dynamic training regimes slightly improve weaker architectures but fail to overcome fundamental architectural limitations.

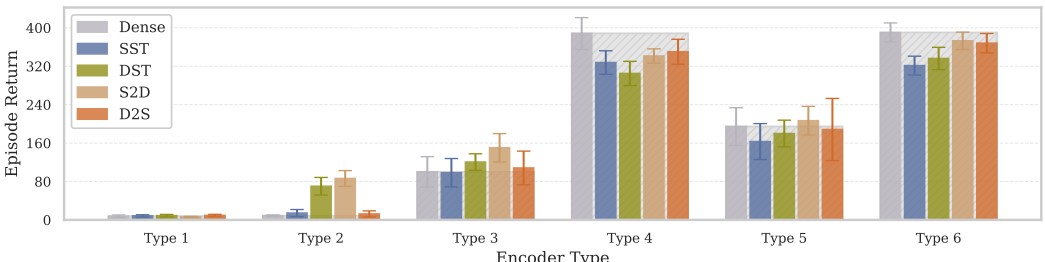

Figure 15: Impact of encoder architecture types and training regimes on agent performance at default size. The results demonstrate that architectural improvements (progression from Type 1 to Type 6) yield substantially greater performance gains than differences between training regimes.

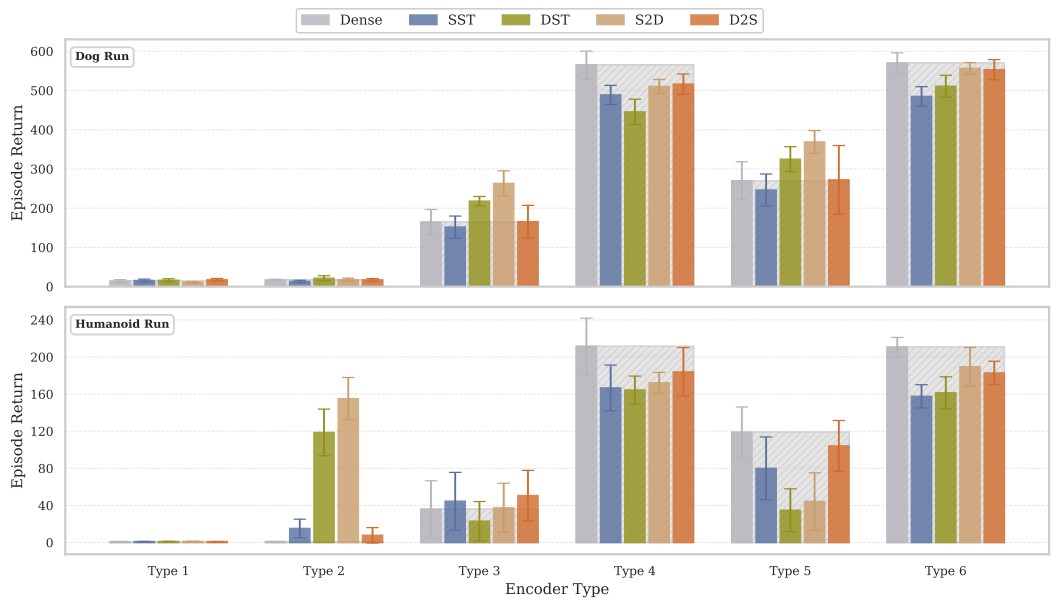

Figure 16: Detailed performance comparison of encoder architecture types (1-6) and training regimes (Dense, SST, DST, S2D, D2S) on Dog Run and Humanoid Run at default network size. Note that architectures with layer normalization (Types 3-6) substantially outperform those without (Types 1-2), regardless of training regime employed.

Figure 16 shows the detailed performance comparison of different encoder architecture types and training regimes at the default model size (approximately 2M parameters) on Dog Run and Humanoid

Run tasks. These results demonstrate that architectural improvements (progression from Type 1 to Type 6) provide substantially greater performance gains than differences between training regimes. Networks without layer normalization (Types 1-2) show consistently poor performance across all training regimes, while networks with both layer normalization and residual connections (Types 5-6) achieve strong performance regardless of training approach.

**Scale Trend.** Figure 17 illustrates how scaling affects performance across representative encoder architectures (Types 2, 4, and 6) and training regimes. As parameter counts increase by factors of 3 and 5, the importance of architectural design becomes even more pronounced. Type 2 networks fail catastrophically at larger scales regardless of training regime, Type 4 networks benefit from sparsity-based approaches when scaled, and Type 6 networks maintain strong performance across all scales with dense training. This progression clearly demonstrates that well-designed architectures create inherently better optimization landscapes that enable effective scaling.

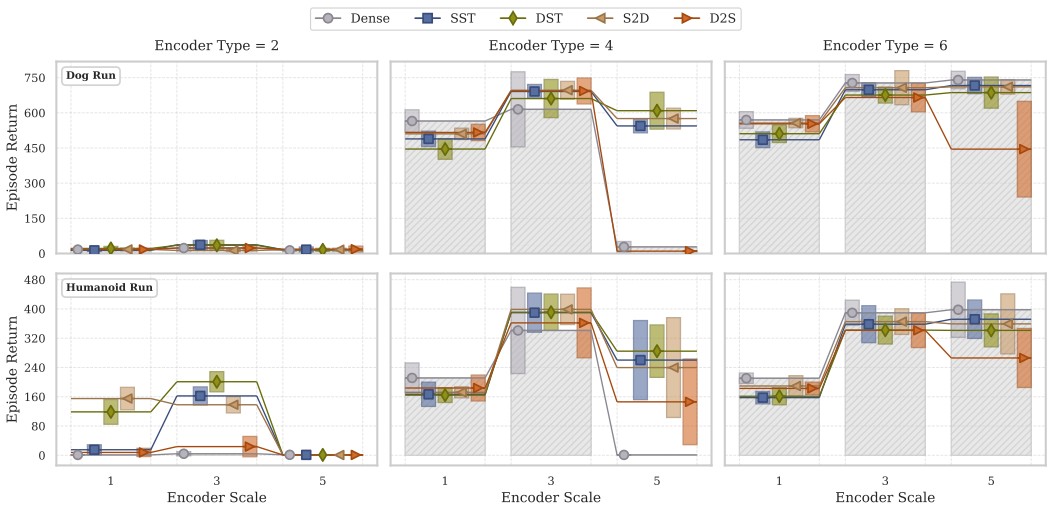

Figure 17: Scaling behavior of encoder performance as parameter count increases across three orders of magnitude (Scale 1: 2M, Scale 3: 37M, Scale 5: 154M) for representative architecture types. Results are shown for Dog Run (top) and Humanoid Run (bottom) tasks. While network sparsity and dynamic training regimes can improve scalability for some architectures, the fundamental network design remains the primary determinant of scaling potential.

## D.2 CRITIC

Following our encoder analysis, we next investigate critic networks that learn through temporal difference (TD) bootstrapping. With a powerful encoder providing stable state representations, we examine how different critic architectures and training regimes impact performance and scalability.

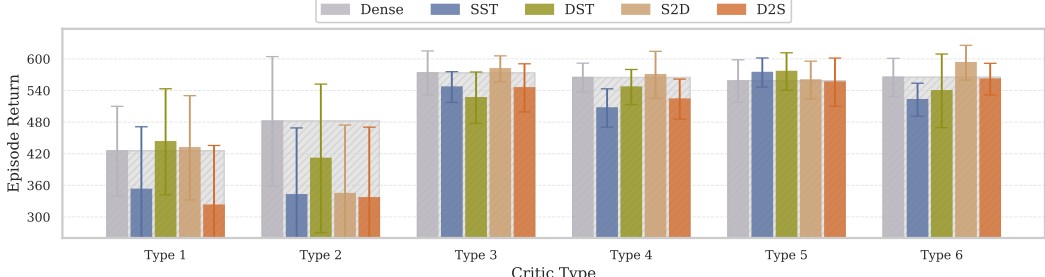

Figure 18: Comparison of critic architecture types and training regimes at default network size. With a strong encoder providing effective state and action representations, critic networks reliably achieve stable optimization. Beyond layer normalization, other architectural advances and training regime variations show limited impact on critic performance at this default scale.

**Default Scale.** Figure 19 presents the performance comparison of different critic architecture types and training regimes at the default model size (approximately 3M parameters). With effective state representations from our encoder, performance differences between architectures and training regimes become less pronounced at this default scale. The most significant improvement comes from layer normalization (comparing Types 1-2 vs. Types 3-6), which provides consistent benefits across all training approaches. This aligns with recent research highlighting layer normalization's critical role in mitigating plasticity loss and reducing value overestimation in TD-based critic training (Nauman et al., 2024a; Lyle et al., 2024a).

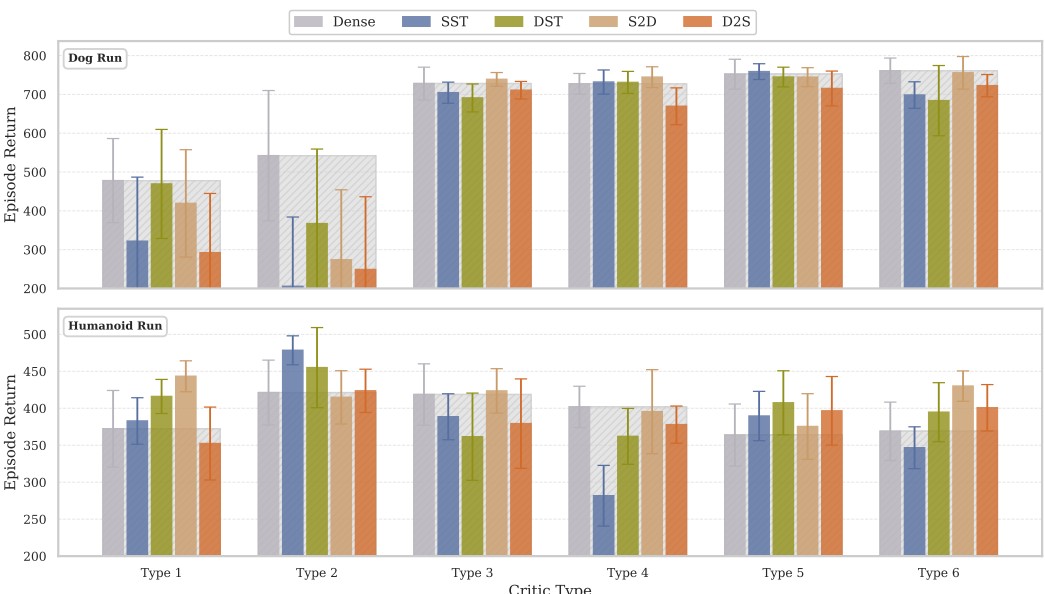

Figure 19: Performance comparison of critic architecture types (1-6) and training regimes at default network size on Dog Run (top) and Humanoid Run (bottom) tasks. With a powerful encoder providing effective state representations, critic networks achieve more consistent performance across architectures compared to encoders. Layer normalization emerges as the most important architectural element for critic networks at default scale.

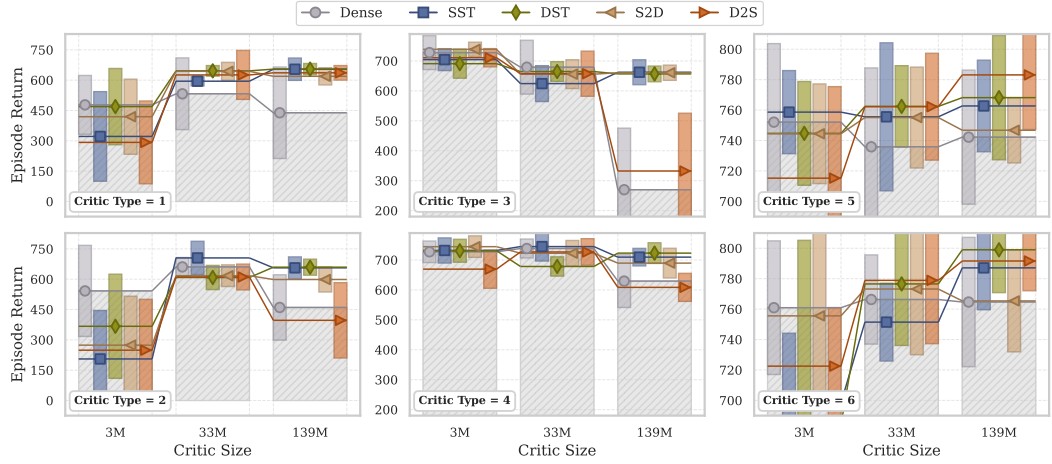

Figure 20: Scalability of critic networks on the Dog Run task across different architecture types and training regimes. As parameter count increases from 3M to 33M and 139M, dense networks (gray) suffer severe performance degradation in architectures without residual connections. Dynamic training approaches, particularly DST (green), consistently provide the best scaling performance. Networks with both advanced architecture (Types 5-6) and dynamic training achieve the highest and most stable results at large scales.

**Scale Trend.** Figure 20 and Figure 21 reveal pronounced differences between architectures and training regimes as critic networks are scaled up. At larger scales (33M and 139M parameters), dense networks suffer severe performance degradation across most architectures, with catastrophic collapse particularly evident in networks without residual connections (Types 1-4). Training regimes that introduce network sparsity and dynamicity substantially enhance critic scalability, with Dynamic Sparse Training (DST) demonstrating the most consistent improvements by maintaining constant sparsity while enabling topology adaptation throughout training.

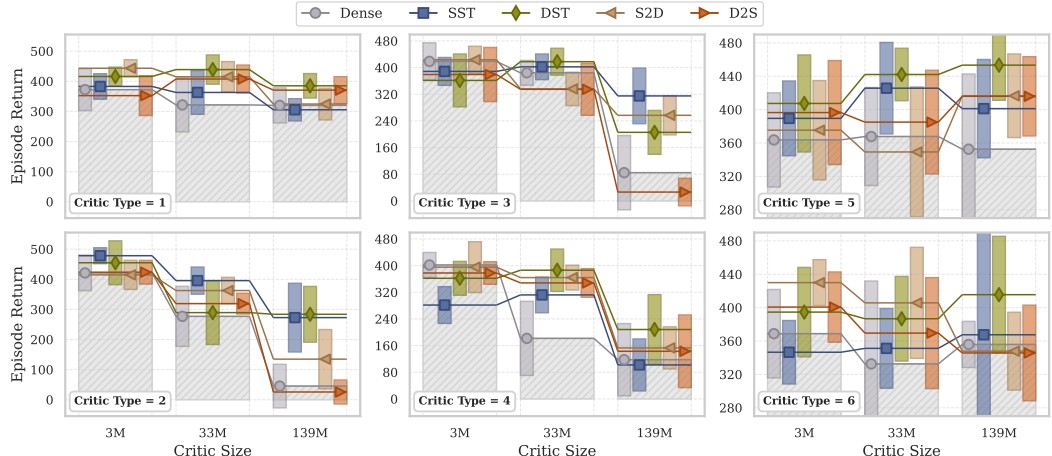

Figure 21: Scalability of critic networks on the Humanoid Run task across architecture types and training regimes. The results parallel those seen on Dog Run, with dense networks showing catastrophic collapse at larger scales, particularly for Types 1-4. Dynamic Sparse Training (DST) consistently enables better scaling across architectures, and the combination of advanced architecture features (Types 5-6) with dynamic training regimes produces the most stable scaling behavior.

These results reveal a unique characteristic of TD-based learning: neither architectural innovations nor training regimes alone can overcome the critic scaling barrier. Achieving full scalability requires both robust architectural foundations and specialized training regimes that induce beneficial network properties.

D.3 ACTOR

Finally, we examine actor networks which directly optimize deterministic policy gradients, creating distinct stability requirements and optimization challenges compared to encoders and critics. Here we maintain a scaled Type 6 encoder (37M parameters) and default Type 4 critic (3M parameters) to isolate actor effects.

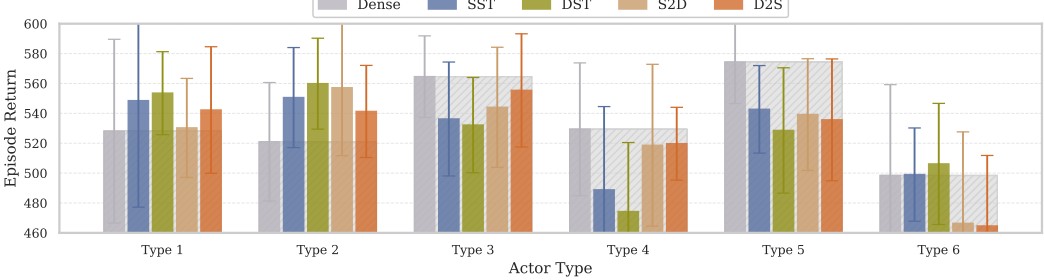

Figure 22: Impact of actor architecture types and training regimes at default network size. Layer normalization provides consistent benefits, while switching activation function from ReLU to ELU significantly reduces performance. Dynamic training approaches help networks without normalization but offer no advantages when applied to more advanced architectures.

**Default Scale.** Figure 23 illustrates the performance of different actor architectures and training regimes at default network size (approximately 1.3M parameters). Layer normalization consistently

benefits all actor architectures regardless of training regime, similar to our observations with critics. However, unlike encoders where ELU activation improved performance, switching from ReLU to ELU significantly degrades actor performance. This suggests that feature sparsity from ReLU better supports policy gradient optimization despite potential dormant neuron concerns. Dynamic training approaches help simpler architectures without normalization but provide no advantages for more advanced configurations, indicating that actors maintain stable optimization pathways with proper architectural elements.

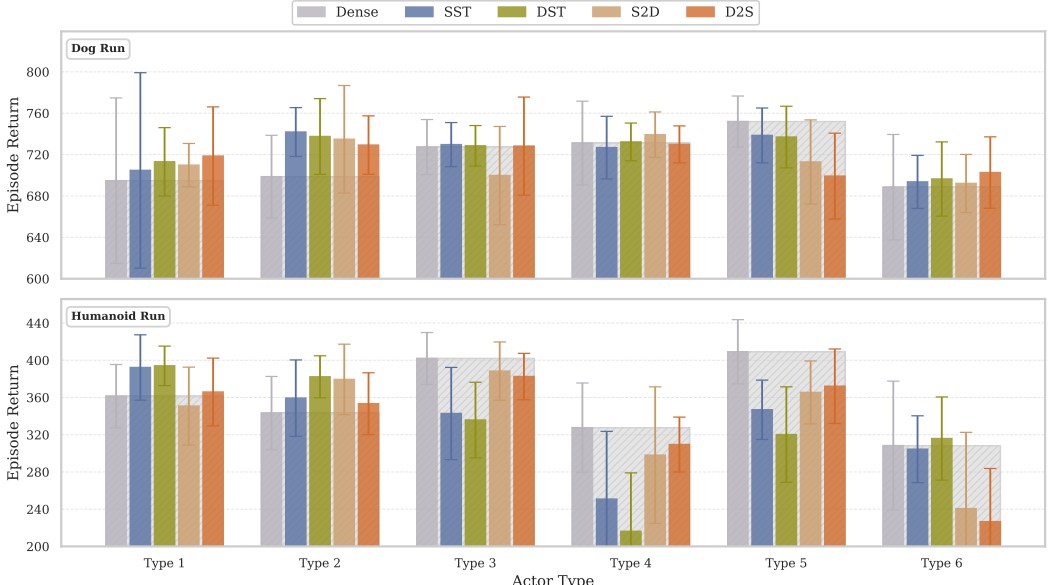

Figure 23: Performance comparison of actor architecture types (1-6) and training regimes at default network size on Dog Run (top) and Humanoid Run (bottom) tasks. Layer normalization provides consistent benefits across all architectures, while ELU activation surprisingly degrades performance compared to ReLU. Dynamic training regimes offer little advantage for actors with well-designed architectures, unlike the pattern observed with critics.

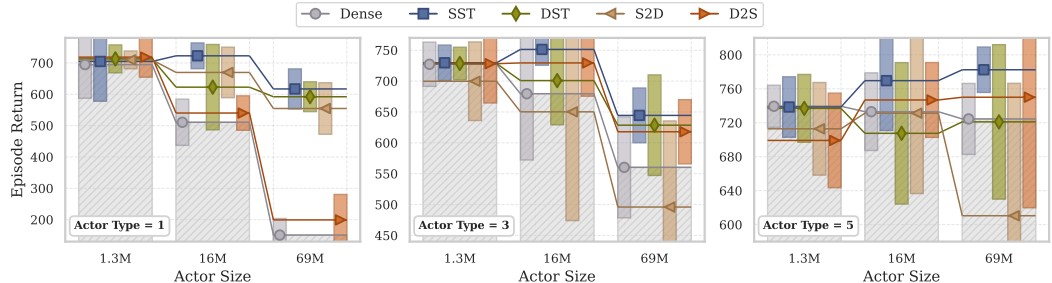

Figure 24: Scaling behavior of actor networks on the Dog Run task as parameter count increases from 1.3M to 16M and 69M. For networks without residual connections (Types 1 and 3), performance collapses dramatically at the largest scale regardless of training regime. Static Sparse Training (SST, blue) provides the most consistent performance gains across scales, contrasting sharply with critics where dynamic approaches worked best.

**Scale Trend.** Figure 24 and Figure 25 reveal clear differences in scaling patterns across actor training regimes. As network size increases to 16M and 69M parameters, normalization and residual connections become essential to prevent catastrophic performance decline. Most strikingly, and contrary to our findings with critics, Static Sparse Training (SST) consistently outperforms all

dynamic approaches for scaled actor networks. This advantage becomes more pronounced at larger scales, where dynamic approaches like DST show much higher performance variance than SST.

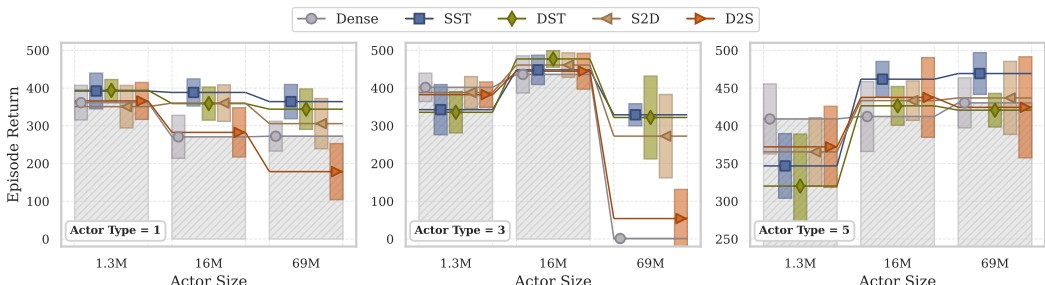

Figure 25: Scaling behavior of actor networks on the Humanoid Run task across different architectures and training regimes. The results mirror the Dog Run findings, with SST providing superior scaling performance. Networks with residual connections (Type 5, right column) maintain performance even at the largest scale when trained with static sparsity. The contrast with critic scaling behavior highlights the fundamentally different optimization requirements of actor and critic networks.

These results highlight a fundamental trade-off in actor networks that differs from critics: while mitigating plasticity loss remains important, preserving stable optimization pathways is equally crucial for policy learning. Static Sparse Training provides this ideal balance by reducing parameter redundancy without disrupting established gradient flows, explaining its superior performance in policy network scaling. This pattern directly contrasts with critics, where dynamic training performed best, emphasizing the need for module-specific training approaches in DRL.

## E    EXTENDED EXPERIMENTS OF MODULE-SPECIFIC TRAINING (MST)

To demonstrate MST's broader applicability beyond MR.Q, we evaluate its effectiveness on MR.SAC, which combines the self-supervised encoder training from MR.Q with SAC's actor-critic learning. We test three scaling approaches: scaling depth only, scaling width only, and scaling both dimensions.

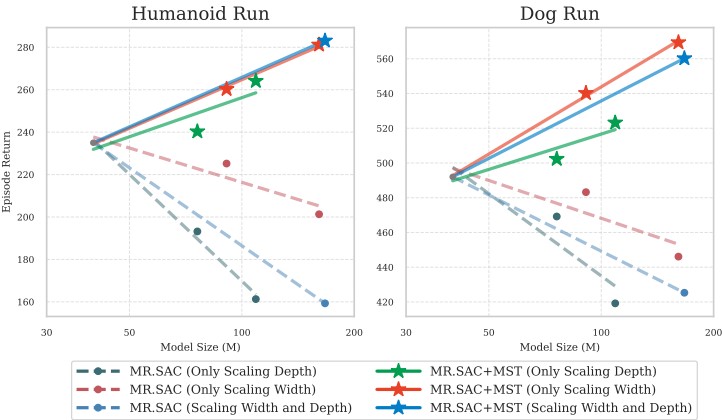

Figure 26: MST enables successful scaling in MR.SAC across different scaling dimensions. Standard scaling approaches (dashed lines) lead to performance degradation as model size increases, while MST (solid lines) successfully harnesses the benefits of scaling. Results averaged over 5 seeds on Humanoid Run and Dog Run tasks.

Figure 26 shows that standard scaling consistently degrades performance across all approaches and both tasks. In contrast, MST enables successful scaling across all dimensions. These results confirm that MST's benefits generalize beyond the specific MR.Q framework to other RL algorithms that incorporate model-based representation learning, establishing its practical utility for scalable DRL.