# OpenReview forum: "Rethinking the Role of Dynamic Sparse Training for Scalable Deep Reinforcement Learning"
_ICLR.cc/2026/Conference — Submitted to ICLR 2026_

### Official Review · Reviewer_hBbY · 2025-10-26

**Soundness:** 2
**Presentation:** 4
**Contribution:** 2
**Rating:** 4
**Confidence:** 5

**Summary:**

The work investigates the interplay between sparsity methods, architectural choices, and scaling behavior in deep RL. A combination of sparsity methods and architectural choices is proposed as a gold standard for achieving a scaling behavior. This combination is identified through a grid search over standard sparsity methods and architectural choices, performed on two DMC tasks using Mr.Q as the base algorithm. Importantly, different combinations are selected for the encoder, the critic, and the actor. Finally, a broader experimentation is performed to evaluate the proposed combinations.

**Strengths:**

I. The method investigates an important topic in RL: performance increase with respect to the model size, i.e., the scaling behavior.

II. The method is clearly presented, the reading flow is good, and sufficient details are disclosed to understand the approach fully.

**Weaknesses:**

A. In its current form, the contribution of the work is limited:

   i. The proposed method is designed from a grid search on only two tasks from the same benchmark using a single algorithm with only one set of hyperparameters. This means the relevance of this work may quickly become outdated as new architectures, RL algorithms, or RL tasks are introduced. Therefore, I suggest focusing more on the conclusions that can be drawn from the results, rather than developing a new algorithm intended to serve as a goal standard for achieving scalability.

   ii. The proposed combination is determined sequentially, which is potentially underperforming compared to the optimal combination. Indeed, the selected combination might differ if the pruning algorithm for the actor is selected before the one for the encoder. This calls into question the conclusions drawn from the results. A discussion about this aspect would clear this problematic point.

   iii. After designing the proposed method from experimental outcomes, a broader analysis is made to evaluate MST on different tasks and algorithms. While MST outperforms the dense approach, it is not shown how a baseline using the same pruning algorithm for the actor, the critic, and the encoder, combined with residual connections and efficient normalization layers, would perform. Adding those baselines would strengthen the claim that a specific pruning algorithm is needed for each module. As this baseline would use less computational budget than MST requires for training, a grid search over different sparsity levels of the shared pruning algorithm can be considered. For now, this analysis is shown only in Figure 11, where the confidence intervals of the baseline using SST for both the actor and the critic strongly overlap with MST, while requiring less computational budget, since MST is obtained via a grid search.

B. The claims are generally too broad, given the fact that the method is purely empirical and evaluated on a limited set of tasks from the same benchmark. I suggest changing the claims to reflect the presented work:

   i. While the following claim is stated in the general context in Line 455: "DST alone cannot match architectural improvements", the submission only demonstrates it on two tasks from the same benchmark for Mr.Q only. Moreover, the sparsity level was not tuned, which invalidates the claim.

   ii. In Line 457, "Second, effective dynamic training demands module-specific approaches", this claim is, for now, not justified as discussed in weakness A. iii.

   iii. The three lessons provided in the conclusion are problematic. First, Lesson 2 is not clearly proven, as discussed in weakness A. iii. Then, Lesson 3 is not linked to the submission's content. It can be used as an opening, but not as a lesson. Finally, the remaining lesson, Lesson 1, is already known by the community. Indeed, as stated, "[the empirical analysis] confirms that DRL optimization pathologies require distinct solutions beyond conventional DL practices.", which means that it cannot be used as a lesson from this work.

C. Some discussions are missing:

   i. The proposed approach is not easy to implement, as every module follows a different pruning algorithm. Discussing in which scenario the approach's benefits would outweigh the required additional implementation/tuning would be helpful for practitioners who would like to leverage this work.

   ii. Discussing ways of adapting the proposed approach to vision-based experiments would be helpful.

   iii. Discussing the influence of observation normalization on the performance of MST would be interesting.

D. Suggestions:

   i. In the first paragraph of Section 5, [1] can be cited a "recent study [that] further revealed that such dynamic training strategies can alleviate the scaling barriers caused by DRL’s unique optimization pathologies"

[1] Vincent, Théo, et al. "Eau De Q-Network: Adaptive Distillation of Neural Networks in Deep Reinforcement Learning." RLJ 2025.

   ii. In the second paragraph of Section 2.2, [1] can be cited as an example of modern activation function.
[1] Delfosse, Q., et. al. Adaptive Rational Activations to Boost Deep Reinforcement Learning. ICLR 2024.

   iii. Increasing the font size in the figures would make them much easier to read.

**Questions:**

1. What is "zsa network", stated in Line 229?

2. In Section 3.3, which pruning method is used by the critic?

---

### Official Review · Reviewer_vJgN · 2025-10-27

**Soundness:** 1
**Presentation:** 2
**Contribution:** 2
**Rating:** 2
**Confidence:** 4

**Summary:**

This paper analyzes the performance of sparse training under different DRL components, architectures, and sparsity strategies, and proposes Module-Specific Training (MST) to enhance the scalability of DRL.

**Strengths:**

This paper investigates how sparse training can enhance the scalability of DRL and presents numerous case studies.

**Weaknesses:**

1. The authors implemented RigL strategy as DST but did not explore other strategies for DST, limiting the analysis.
2. The network variants analyzed are insufficient. More activation functions should be analyzed. Similarly, different types of convolutional networks, normalization methods, and network regularization techniques should also be examined.
3. The experiments mainly compare the return metric, but lack comparisons on hardware metrics, such as wall-clock throughput, FLOPs, and VRAM usage. These metrics should be compared between the proposed MST method and original DRL methods.
4. Why is the sparsity set to 0.6? How does the performance of MST change with different sparsity levels, such as 0.8, 0.9, or 0.95?
5. This paper investigates the sparse or dense configuration strategies for the encoder, actor, and critic components in DRL, but it is unclear how sparsity configurations should be applied to other types of components, such as world models. Additionally, when DRL has multiple actors or critics, for example in evolutionary DRL, whether each actor or critic should use the same sparsity configuration is not explored.
6. When the proposed method is applied to multi-agent DRL, will each agent use the same sparse/dense configuration?
7. The paper assumes the same sparsity of 0.6 for all components. Should the actor and critic use different sparsity levels? If there are multiple actors or critics, should each network use different sparsity levels?
8. The proposed MST method is based on empirical findings and lacks theoretical analysis. It is recommended that the authors provide a theoretical explanation for the effectiveness of MST.
9. The validity of the proposed method is only verified on SAC and DDPG. It is suggested to test the method on at least five other SOTA DRL methods. Furthermore, verification should be conducted on various DRL tasks, such as multiple MuJoCo tasks and robotic manipulation tasks.

**Questions:**

1. The authors implemented RigL strategy as DST but did not explore other strategies for DST, limiting the analysis.
2. The network variants analyzed are insufficient. More activation functions should be analyzed. Similarly, different types of convolutional networks, normalization methods, and network regularization techniques should also be examined.
3. The experiments mainly compare the return metric, but lack comparisons on hardware metrics, such as wall-clock throughput, FLOPs, and VRAM usage. These metrics should be compared between the proposed MST method and original DRL methods.
4. Why is the sparsity set to 0.6? How does the performance of MST change with different sparsity levels, such as 0.8, 0.9, or 0.95?
5. This paper investigates the sparse or dense configuration strategies for the encoder, actor, and critic components in DRL, but it is unclear how sparsity configurations should be applied to other types of components, such as world models. Additionally, when DRL has multiple actors or critics, for example in evolutionary DRL, whether each actor or critic should use the same sparsity configuration is not explored.
6. When the proposed method is applied to multi-agent DRL, will each agent use the same sparse/dense configuration?
7. The paper assumes the same sparsity of 0.6 for all components. Should the actor and critic use different sparsity levels? If there are multiple actors or critics, should each network use different sparsity levels?
8. The proposed MST method is based on empirical findings and lacks theoretical analysis. It is recommended that the authors provide a theoretical explanation for the effectiveness of MST.
9. The validity of the proposed method is only verified on SAC and DDPG. It is suggested to test the method on at least five other SOTA DRL methods. Furthermore, verification should be conducted on various DRL tasks, such as multiple MuJoCo tasks and robotic manipulation tasks.

---

### Official Review · Reviewer_34s7 · 2025-10-27

**Soundness:** 2
**Presentation:** 3
**Contribution:** 2
**Rating:** 4
**Confidence:** 4

**Summary:**

This paper investigates the role of dynamic sparse training (DST) in scaling deep reinforcement learning (DRL) networks. The authors conduct a systematic empirical study examining how different training regimes (dense, static sparse, dynamic sparse, sparse-to-dense, dense-to-sparse) interact with architectural improvements across three DRL modules: encoder, critic, and actor. Based on their findings, they propose Module-Specific Training (MST), which assigns tailored training strategies to each module: dense training for encoders, DST for critics, and static sparse training (SST) for actors.

**Strengths:**

1. **Well-motivated research question.** The paper addresses an important gap in understanding how dynamic sparse training interacts with architectural improvements for DRL scalability, moving beyond treating them as independent solutions.
2. **Systematic experimental design.** The module-specific investigation is well-structured, examining encoder, critic, and actor separately with consistent experimental protocols across 6 architectural variants and 5 training regimes.
3. **Clear practical guidance.** MST offers actionable recommendations that can be applied to existing DRL algorithms without algorithmic modifications, demonstrated on SAC, DDPG, and MR.Q/MR.SAC.

**Weaknesses:**

1. **Severely limited scope of DST strategies.** The paper only implements RigL for DST while claiming to provide "a comprehensive comparison between different dynamic approaches." Other well-established DST methods, such as SET variants, SNFS, or gradient flow-based approaches, are entirely ignored. This fundamentally undermines the generalizability of conclusions about the effectiveness of DST.

2. **Lack of computational efficiency analysis.** The paper completely omits critical metrics, including wall-clock training time comparisons between dense and sparse methods, FLOPs/throughput measurements, Memory consumption (VRAM usage), and actual training costs. ** ** ** This is particularly problematic given that computational efficiency is a primary motivation for sparse training in the broader literature.

3. **Arbitrary and unjustified hyperparameter choices.** Sparsity fixed at 0.6 with only a brief mention in Section 4 ablation, no systematic study of 0.7, 0.8, 0.9, 0.95 as commonly explored in sparse training literature. No justification for ER initialization vs uniform beyond citing prior work. RigL hyperparameters (update frequency, drop fraction schedule) appear copied from vision tasks without DRL-specific tuning.

4. **Theoretical understanding completely absent.**: The paper is purely empirical without any theoretical analysis of: Why do different modules respond differently to sparsity/dynamicity? What properties of TD learning, policy gradients, and self-supervised learning create these differences? Under what conditions would MST recommendations hold vs fail? Connection to optimization landscape theory or neural tangent kernel perspectives?

**Questions:**

Please see weaknesses.

---

### Official Review · Reviewer_AWvW · 2025-10-28

**Soundness:** 3
**Presentation:** 4
**Contribution:** 3
**Rating:** 6
**Confidence:** 3

**Summary:**

The manuscript presents a solid summary and comprehensive analysis concerning different techniques that introduce network dynamicality and sparsity for network scalability, and proposes a module-specific training framework for DRL. The three drawbacks are worth investigating, the investigation is promising, and provides solid ground for the proposed module-specific framework. While the proposed method is not essentially novel, the conclusion is somewhat valuable to motivate future fine-grained model design in sparse DRL.

**Strengths:**

1. Comprehensive analysis and investigation
2. Easy-to-follow model design

**Weaknesses:**

1. Limited novelty of the approach
2. Somewhat straightforward but trivial discovery of the module-specific impact on the model scalability

**Questions:**

1. While I agree that the investigation of the impact from different components is necessary and informative for future model design, do the authors consider comparing only applying sparse techniques on a single component with the traditional setting of applying sparse techniques uniformly across all modules? Is it possible that the application of sparse techniques in multiple components together can have different impacts, like counteraction?

2. It seems that the authors directly apply the theoretical optimal techniques for different modules in Section 4, without considering their possible interaction. Is it possible that the combination of techniques used will have unexpected results?

3. Given that all investigation is conducted based on MR.Q framework, is it possible that the results and conclusions are limited to specific encoder, actor, critic, etc.? In other words, from the current presentation, the takeaway remarks at the end of each subsection in Section 3 may need more support and comprehensive experiments.

---

### Meta-Review · Area_Chair_3Jep · 2026-01-07

**Summary:**

This manuscript presents a systematic empirical investigation of dynamic sparse training (DST) for scalable deep reinforcement learning, examining how sparsity and dynamicity interact with architectural improvements across encoders, critics, and actors. Based on these observations, the authors propose Module-Specific Training (MST), a framework that assigns different sparse/dynamic training strategies to different network modules. Reviewers agreed that the paper is well motivated and addresses an important question in reinforcement learning scalability.

However, reviewers raised multiple concerns regarding the narrow coverage of DST methods (largely limited to RigL), the absence of computational efficiency analysis (e.g., wall-clock time, FLOPs, memory), the restricted range of tasks and architectures considered, and the breadth of claims relative to the evidence presented. These issues could not be clarified or addressed, as no rebuttal was provided.

**Reviewer Concerns:**

Reviewers consistently noted limited novelty, reliance on a single DST method, lack of efficiency analysis, narrow experimental scope, and absence of theoretical justification; these concerns remained unaddressed due to the lack of a rebuttal.

**Reviewer Scores:**

In the absence of a rebuttal upward score revision would be unlikely.

---

### Decision · Program_Chairs · 2026-01-26

Reject